# Towards a Formal Theory of Representational Compositionality

**Eric Elmoznino** [* 1 2]   **Thomas Jiralerspong** [* 1 2]   **Yoshua Bengio** [1 2]   **Guillaume Lajoie** [1 2]

## Abstract

Compositionality is believed to be fundamental to intelligence. In humans, it underlies the structure of thought and language. In AI, it enables a powerful form of out-of-distribution generalization, in which a model systematically adapts to novel combinations of known concepts. However, while we have strong intuitions about what compositionality is, we lack satisfying formal definitions for it. Here, we propose such a definition called *representational compositionality* that is conceptually simple, quantitative, and grounded in algorithmic information theory. Intuitively, representational compositionality states that a compositional representation is both expressive and describable as a simple function of parts. We validate our definition on both real and synthetic data, and show how it unifies disparate intuitions from across the literature in both AI and cognitive science. We hope that our definition can inspire the design of novel, theoretically-driven models that better capture the mechanisms of compositional thought. We make our code available here.

## 1. Introduction

Compositionality is thought to be one of the hallmarks of human cognition. In the domain of language, it lets us produce and understand utterances that we have never heard, giving us "infinite use of finite means" (Chomsky, 1956). Beyond this, one of the most influential ideas in cognitive science is the *Language of Thought* hypothesis (Fodor, 1975; Quilty-Dunn et al., 2023), which conjectures that *all* thought involved in higher-level human cognition is compositional. Compositionality has been equally influential in AI from its very origins, motivating efforts in neurosymbolic AI (Garcez & Lamb, 2023; Sheth et al., 2023; Marcus, 2003),

---
[*]Equal contribution   [1]Mila – Quebec AI Institute [2]Université de Montréal. Correspondence to: Eric Elmoznino <eric.elmoznino@mila.quebec>, Guillaume Lajoie <guillaume.lajoie@mila.quebec>.

*Proceedings of the 42nd International Conference on Machine Learning*, Vancouver, Canada. PMLR 267, 2025. Copyright 2025 by the author(s).

probabilistic program inference (Lake et al., 2017; Ellis et al., 2023), modular deep neural networks Bengio (2017); Goyal & Bengio (2022); Pfeiffer et al. (2023); Andreas et al. (2016); Goyal et al. (2021; 2020); Schug et al. (2024), disentangled representation learning (Higgins et al., 2017; Lachapelle et al., 2022; Ahuja et al., 2022; Brehmer et al., 2022; Lippe et al., 2022; Sawada, 2018), object-centric learning (Locatello et al., 2020; Singh et al., 2023; Wu et al., 2024), and chain-of-thought reasoning (Wei et al., 2022; Kojima et al., 2022; Hu et al., 2024), to name only a few. One of the primary appeals of compositionality is that it enables a powerful form of out-of-distribution generalization (Lake & Baroni, 2018): if a model is compositional with respect to a set of features in its training data, it need not observe all possible combinations of those features in order to generalize to novel ones (Schug et al., 2024; Wiedemer et al., 2024; 2023; Bahdanau et al., 2019; Mittal et al., 2021).

Despite its importance, compositionality remains an elusive concept: there are currently few formal, quantitative definition of compositionality that could be used to measure it, and those that do exist have important theoretical drawbacks. Compositionality is often described in the following way:

**Definition 1 (*Compositionality – colloquial*)**

> *The meaning of a complex expression is determined by its structure and the meanings of its constituents (Szabó, 2022).*

In the context of neural representations in brains or deep neural networks (DNNs), we can take these "meanings" to be high-dimensional vectors of activations. Definition 1 lacks formal rigour and breaks down upon inspection.

First, the definition presupposes the existence of a symbolic "complex expression" associated to each meaning. In some cases, this makes sense; for instance, consider human languages and the neural representations they elicit. But where do these expressions and their constituent parts come from when considering neural representations themselves such as in the Language of Thought hypothesis, where thoughts are encoded in distributed patterns of neural activity?

Second, it is unclear what the expression's "structure" should be. The definition is motivated from human language, where sentences have syntactic parses and individual words have types (e.g., noun, verb), but these properties are not intrinsic to the sentences themselves, which are simply strings.

Third, the definition says that meaning is "determined by" the structure and meanings of the constituents, but it does not put any kind of restriction on these semantics for the meanings to qualify as compositional: any function qualifies. For instance, functions that *arbitrarily* map constituents to their meanings (as in the case of idioms like "he kicked the bucket") are functions nonetheless and thus satisfy Definition 1, but it is commonly agreed that they are not compositional (Weinreich, 1969; Mabruroh, 2015; Swinney & Cutler, 1979).

Finally, the colloquial definition of compositionality suggests that it is a binary property of representations, when it should arguably be a matter of degree. For instance, while linguists often model the syntax and semantics of language using hierarchical decompositions that are considered compositional (Chomsky, 1956), human language regularly deviates from this idealization. In particular, language has some degree of context-sensitivity, where the meanings of words depend on those of others in the sentence. Thus, human language does not satisfy the colloquial binary definition of compositionality, even though it is considered largely compositional.

The colloquial definition of compositionality is thus flawed if we wish to formalize and measure it quantitatively, moving beyond mere intuitions that are fundamentally limited in their explanatory reach. In this paper, we introduce such a definition, which we call *representational compositionality*. The definition is grounded in algorithmic information theory, and says that compositional representations are both expressive and easily describable as a simple function of symbolic parts. We argue that this definition not only addresses Definition 1's flaws, but also accounts for and generalizes our many intuitions about compositionality. Finally, we provide empirical experiments that clarify implications of the definition and validate its agreement with intuition. Since representational compositionality is rigorous and quantitative, we hope that it can inspire new principled methods in AI for learning compositional representations.

## 2. Compressing a Representation

The definition that we will propose rests on the idea that compositional representations can be described as a simple function of constituent parts. While there may be many ways to describe a representation, a natural and principled way is through the lens of *optimal compression* and Kolmogorov complexity. We provide a brief introduction to Kolmogorov complexity below, and give more background Appendix A.

**Kolmogorov complexity**  Kolmogorov complexity (Li et al., 2008; Kolmogorov, 1965) is a notion of information quantity. Intuitively, the Kolmogorov complexity of an object $x$, denoted $K(x)$, is the length of the shortest program (in some language) that outputs $x$. A related notion is the conditional Kolmogorov complexity of $x$ given another

object $y$, denoted $K(x|y)$, which is the length of the shortest program that takes $y$ as input and outputs $x$. Kolmogorov complexity has many intuitive properties as a measure of information quantity. The more "structure" an object has (regularity, patterns, rules, etc.), the more it can be *compressed* using a short program.

In the context of ML, an interesting quantity is the Kolmogorov complexity of a dataset $X = (x_1, ..., x_n)$ where each sample is drawn from a distribution $p(x)$. If the dataset is sufficiently large, the optimal method for compressing it is to first specify $p$ and then encode the data using it, giving $K(X) = K(X|p) + K(p)$ (Fortnow, 2000). For the first term $K(X|p)$, each sample can be optimally encoded using $-\log_2 p(x_i)$ bits, as in the case of Shannon information (Shannon, 2001). The second term $K(p)$ refers to the complexity of the data distribution (i.e., the length of the shortest program that computes the function $p: \mathcal{X} \to \mathbb{R}^+$).

**Compressing $Z$ as a function of parts**  Let us denote a representation by a matrix $Z \in \mathbb{R}^{N \times D}$, where each row $z_n$ is obtained by sampling *iid* from some data distribution and model $p(x)p(z|x)$. For instance, $p(x)$ could be a distribution over natural images, $z_n \sim p(z|x)$ could be the (often deterministic) output of some intermediate layer in a trained image classifier, and the resulting representation $Z \in \mathbb{R}^{N \times D}$ would be a matrix of these layer activations. We will argue that a natural way to think about compositional representations is that they can be significantly compressed as a function of constituent parts. In other words, the shortest program that outputs the representation, with length $K(Z)$, has a particular form: it first describes $Z$ using short parts-based constituents, and then maps these parts to the high-dimensional representation. This program form is shown in Figure 1 and described in detail below. We also give a summary of all program components in Table 1. Crucially, the components of this program will be used in Section 3 to construct our formal definition of compositionality, in which representations that are *more compressible* as a function of constituent parts are *more compositional*. Before combining them into a definition of compositionality, we now describe the components of this program in the following steps.

**Step 1: describe $Z$ using parts-based constituents**  We assume that every sample of the representation $z_n$ of data point $x_n$ can be compressed using a sequence of constituent parts, which in practice are discrete tokens. By analogy to natural language, we will call these sequences "sentences", denoted by $W \in \mathcal{V}^{N \times M}$ where $\mathcal{V}$ is the vocabulary and $M$ is the maximum sentence length. Each row in $W$ is a sentence that describes a high-dimensional vector in the corresponding row of $Z$. Importantly, these are not sentences in a human language like English: they are sequences of discrete tokens that best compress the representation, and can be thought of as the representation's intrinsic language. For instance, if the repre-

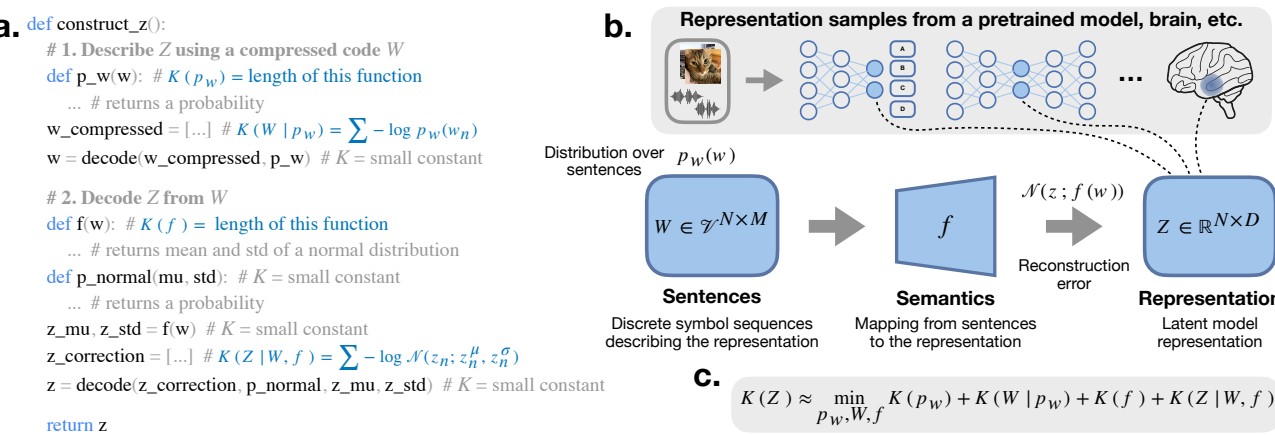

**a.**
```
def construct_z():
    # 1. Describe Z using a compressed code W
    def p_w(w):  # K ( p_w ) = length of this function
        ... # returns a probability
    w_compressed = [...]  # K ( W | p_w ) = ∑ − log p_w(w_n)
    w = decode(w_compressed, p_w)  # K = small constant

    # 2. Decode Z from W
    def f(w):  # K ( f ) = length of this function
        ... # returns mean and std of a normal distribution
    def p_normal(mu, std):  # K = small constant
        ... # returns a probability
    z_mu, z_std = f(w)  # K = small constant
    z_correction = [...]  # K ( Z | W, f ) = ∑ − log 𝒩(z_n; z_n^μ, z_n^σ)
    z = decode(z_correction, p_normal, z_mu, z_std)  # K = small constant

    return z
```

**c.** $$K(Z) \approx \min_{p_w, W, f} K(p_w) + K(W \mid p_w) + K(f) + K(Z \mid W, f)$$

*Figure 1.* **Form of the shortest program outputting a compositional representation $Z$. a.** Pseudocode of the program, which describes the representation using sentences $W$ (sequences of discrete tokens) that are compressed using a prior $p_w(w)$, and then maps these sentences to high-dimensional vectors in representation-space using a function $f(w)$ that outputs the sufficient statistics of a Normal distribution. `decode()` is a short function that decodes an object compressed using arithmetic coding (Witten et al., 1987). **b.** Illustration of the program compressing a representation from a pretrained model layer, brain region, etc. **c.** The total Kolmogorov complexity of the representation is estimated by the length of the shortest program that has this form.

| Name | Symbol | Example (for representations of scene images) |
|---|---|---|
| Representation | $Z \in \mathbb{R}^{N \times D}$ | Layer activations of a CNN in response to $N$ scene images |
| Sentences | $W \in \mathcal{V}^{N \times M}$ | Symbol sequence expressing a scene graph for each $z \in Z$ |
| Language | $p_w$ | Distribution over sentences expressing scene graphs |
| Semantics | $f$ | Embed & concatenate each object/relation in the scene graph |
| Recon. error | $\mathcal{N}(z; f(w))$ | Correct remaining error unaccounted for by the semantics |

*Table 1.* Components of assumed shortest program that outputs a compositional representation $Z$

sentation describes visual scenes, the sentences might specify the objects and relations that the scene is composed of.

To encode sentences in their most compressed form, the program should also define a distribution over the sentences $p_w(w)$. This is because optimal coding schemes (e.g., arithmetic coding Witten et al., 1987) allow us to encode an object using only $-\log p(x)$ bits if $p$ is known (see Equation (7)). Together then, the part of the program in Figure 1 that describes a representation using discrete sentences contributes a total Kolmogorov complexity of:

$$K(p_w) + K(W|p_w) = K(p_w) - \sum_{n=1}^{N} \log p_w(w_n).$$

**Step 2: decode $Z$ from $W$** Given sentences $W$, the program must output $Z$. It must therefore define a function $f : \mathcal{V}^M \to \mathbb{R}^D$—which we call the *semantics* in analogy to natural language—that maps discrete tokens sequences to their high-dimensional vector representations. In general, $f(w_n)$ will not perfectly reconstruct $z_n$ ($w_n$ is discrete but $z_n$ is continuous), and these errors must be corrected. This can be achieved if $f$ outputs the sufficient statistics of some distribution in $\mathbb{R}^D$, in which case the number of bits needed

to encode $z_n$ is $-\log p(z_n; f(w_n))$. For simplicity, we take $p$ to be a Normal distribution $\mathcal{N}$ whose mean and standard deviation are given by $f(w_n)$. In sum, the part of the program in Figure 1 that decodes representations from their sentences contributes a total Kolmogorov complexity of:

$$K(f) + K(Z|W,f) = K(f) - \sum_{n=1}^{N} \log \mathcal{N}(z_n; f(w_n)).$$

As a small technical note, because $Z$ lives in a continuous space, it would take an infinite number of bits to encode. Thus, in practice, $Z$ must be discretized to some finite precision and a discrete approximation of the Normal distribution can be used (e.g., the Skellam distribution).

**Summary** The steps above describe a program outputting $Z$. We take representations to be compositional if they are highly compressible as a function of constituent parts (justified in Section 3). Under this framework, the total Kolmogorov complexity of the representation decomposes as:

$$K(Z) = \min_{p_w, W, f} K(p_w) + K(W|p_w) + K(f) + K(Z|W,f)$$

$$(1)$$

The minimization above is important: the shortest program is the one in which $p_w$, $W$, and $f$ are jointly selected so as to minimize the total program length. With $K(Z)$ defined, we can provide some more intuition for its components.

$K(p_w)$ is the complexity of the language used to describe the representation. For instance, a language in which each word is independent of the others would be simpler than a language in which each word is highly context-sensitive. $K(W|p_w)$ is the complexity of the sentences needed to describe the representation using the language $p_w$. If sentences tend to be typical utterances with high probability under the language, they will have low complexity. $K(f)$ is the complexity of the semantics that define how sentences (discrete token sequences) map to their meanings (high-dimensional vectors). This term is central to the definition of compositionality that we will introduce in Section 3. $K(Z|W,f)$ arises from imperfect reconstructions of $Z$, such as errors due to continuous parts of $Z$ that can't be modeled as a function of discrete inputs.

## 3. Representational Compositionality

Our definition of compositionality is a ratio of constituent terms in the decomposition of $K(Z)$ in Equation (1):

**Definition 2 (*Representational compositionality*)**

> *The compositionality of a representation $C(Z)$ is:*
>
> $$C(Z) = \frac{K(Z)}{K(Z|W)} = \frac{K(Z)}{K(f) + K(Z|W,f)}, \qquad (2)$$
>
> *where $W$, and $f$ are obtained from the shortest program that compresses $Z$ in Equation (1).*

Crucially, $p_w$, $W$, and $f$ are *not* free parameters: they are intrinsic to the representation in that they best compress $Z$ (see the minimization in Equation (1)). Like Kolmogorov complexity, $C(Z)$ is intractable to compute, but it can still be tractably estimated using efficient compression and optimization methods. In Appendix B, we outline a strategy for estimating $(p_w, W, f)$ and thus, $C(Z)$ for general cases, though we leave precise implementation for future work. Importantly, Definition 2 can also be adapted for *language systems* where a symbolic mapping $W$ (a language) is given and fixed:

**Definition 3 (*Language system compositionality*)**

> *The compositionality of a language $C^L(Z)$ is:*
>
> $$C^L(Z) = \frac{K(Z)}{K(Z|W^L)} = \frac{K(Z)}{K(f^L) + K(Z|W^L, f^L)}, \qquad (3)$$
>
> *where $f^L$ is obtained from the shortest program that compresses $Z$ given sentences $W^L$.*

Intuitively, $C(Z)$ measures how expressive a representation is relative to how well it can be compressed as a simple function of parts $W$ that constitute an intrinsic language. In $C^L(Z)$, however, the language $W^L$ is externally defined; for instance in a natural language, $W^L$ are the sentences that a person might utter while $Z$ are the neural activity patterns that those sentences elicit. While $C(Z)$ is more general, $C^L(Z)$ can be used to estimate the compositionalities of real-world languages or other mappings between symbolic sequences and representations (e.g. tokenization schemes), making it a useful tool for AI development. In addition, $C^L(Z)$ can be used to investigate a learned representation's compositionality with respect to a dataset's underlying generative factors, as is done in prior work (Ren & Sutherland, 2024; Wiedemer et al., 2023; Ren et al., 2023).

In Section 4, we will first illustrate and test properties of $C(Z)$ using synthetic data where computations are tractable. We follow with real-world evaluation of $C^L(Z)$ on various language systems. Before doing so, we begin by unpacking our definitions to see how they account for the problems of the colloquial Definition 1 and explain computational properties typically associated with compositionality. We discuss related work around compositionality in Section 5 and limitations of our definition in Section 6.

**Expressivity and compression** Representational compositionality says that the compositionality of a representation is a compression ratio that depends on two things: (1) the complexity of the representation (numerator), and (2) the complexity of the semantics which construct the representation from its constituent parts (denominator). When a representation is highly expressive (high $K(Z)$) but can nevertheless be compressed as a *simple* function of constituent parts (low $K(Z|W)$), representational compositionality says that the representation is highly compositional. Representational compositionality therefore formalizes a hypothesis in cognitive science that compositionality emerges from competing pressures for expressivity and compression (e.g., Kirby, 1999; Kirby et al., 2004; 2008, and references therein), which has also recently been explored in AI from a theoretical perspective (Ren & Sutherland, 2024).

**Constituent "parts" are intrinsic to $Z$** Unlike the colloquial Definition 1, representational compositionality makes it clear where the constituent parts (tokens in $W$), complex expressions ($W$), and structure ($f$) associated with a representation come from: optimal compression. This is a significant difference between Definition 2 and other related ideas in the literature which quantify compositionality in terms of reconstruction from *externally*-defined parts (e.g., Andreas, 2019; Trager et al., 2023; Lewis et al., 2022).

**Systematicity and generalization** Representational compositionality formalizes the intuition that the constituent parts

of a compositional representation determine the meaning of the whole in a *systematic* way (Szabó, 2022; 2012), where "systematicity" is a term from cognitive science that roughly means "structured" or "non-arbitrary". If $f$ arbitrarily maps sentences $w$ to their representations $z$ in a way that does not take the structure or words of the sentence into account (as in the case of idioms), then its complexity $K(f)$ is necessarily high and compositionality is low (we demonstrate this through experiments in Section 4.1). In addition, if $f$ is inaccurate in how it maps sentences to their representations, the error $K(Z|W,f)$ is high and the compositionality low. A representation that is highly compositional according to our definition thus benefits from the generalization ability of simple functions (low $K(f)$) that fit their data well (low $K(Z|W,f)$). This ability of $f$ to generalize to novel sentences explains the fundamental relationship between compositionality and notions of systematicity from cognitive science (Szabó, 2022).

**Structure-preserving maps**  Representational compositionality explains the widely-held intuition that semantics functions $f$ which are compositional are structure-preserving in how they map $w \rightarrow z$ (Montague et al., 1970). In a structure-preserving map, each word in the sentence $w$ independently affects a different subspace of the representation $z$ so that pairwise-distances are similar in sentence-space and representation-space. As explained in Ren et al. (2023), structure-preserving maps have lower complexity, and thus higher compositionality according to our definition. We support this claim empirically through experiments in Section 4.1.

**Modularity**  Representational compositionality explains the relationship between compositionality and modularity, which has been difficult to formally articulate in past work (Lepori et al., 2023; Goyal & Bengio, 2022; Mittal et al., 2022). Modularity refers to a system which can be decomposed into interacting sub-parts that can be understood separately (Poole & Mackworth, 2010); an example in ML is mixture-of-experts models. A modular $f$ is simple because it decomposes knowledge into smaller reusable components, each of which only needs to be defined once. This also explains why natural language is highly compositional. Linguists model language using context-free grammars (Chomsky, 1956), in which a sentence decomposes into a parse tree with a "production rule" recursively applied at each node. The production rules are akin to a small number of reusable modules in $f$, giving language simple semantics. We support these claims empirically through experiments in Section 4.1.

Ultimately, a formal definition of compositionality should be judged based on whether it agrees with our intuitions and generalizes them in meaningful ways. Based on the properties listed above, we argue that representational compositionality satisfies all of these desiderata. To provide further intuition for representational compositionality and its implications, we describe some concrete illustrative examples in Appendix D.

## 4. Empirical Results

In this section, we evaluate our compositionality definitions, $C(Z)$ and $C^L(Z)$, on both synthetic and real-world datasets to see if they agree with intuitions.

We compare to a heuristic metric of compositionality called *topological similarity* that is commonly used in the literature. For some $(W,Z)$, topological similarity computes a distance between all pairs of sentences $\Delta_{\mathcal{W}}^{ij} = d_{\mathcal{W}}(w_i, w_j)$ using a distance metric $d_{\mathcal{W}}(\cdot)$ in $\mathcal{W}$, and a distance between all pairs of representation elements $\Delta_{\mathcal{Z}}^{ij} = d_{\mathcal{Z}}(z_i, z_j)$ using a distance metric $d_{\mathcal{Z}}(\cdot)$ in $\mathcal{Z}$. It then computes the Pearson correlation $\rho$ between the two pairwise distance matrices, quantifying the degree to which the two spaces share linear structure.

### 4.1. Synthetic Representations

We first consider representations $Z$ that are generated synthetically using known rules through: $z \sim \mathcal{N}(z; f(w)), w \sim p_w(w)$. Since we know the underlying programs that generated the representations in this case, we know the true complexity terms $K(p_w)$, $K(W|p_w)$, $K(f)$, and $K(Z|W,f)$ needed to compute $C(Z)$ exactly, allowing us to better validate representational compositionality. We describe our synthetic representations below (details in Appendix I).

**Lookup tables**  The simplest way to construct a representation from sequences of discrete tokens is to assign each token in the vocabulary a fixed embedding in a lookup table, and then concatenate these embeddings across the sequence (Figure 2a). Alternatively, the lookup table could assign each unique $n$-gram an embedding and we could concatenate the embeddings for consecutive $n$-sized chunks in the sequence. We call $n$ the "disentanglement" factor because $n = 1$ corresponds to a representation in which each word fully determines a subset of dimensions in the representation. We generate representations by varying certain parameters of the generative program while keeping others constant, and observe the effects on compositionality in Figure 2b.

*Sentence length:* As sentence length increases, compositionality should intuitively increase. For instance, if sentences are of length 1, we are not tempted to call the representation compositional. The more the representation decomposes according to parts, the more compositional it should be. Representational compositionality empirically matches this intuition because $K(Z)$ increases with sentence length (there are more possible $z$ values, for instance) and $K(f)$—proportional to the size of the lookup table—decreases with sentence length (embeddings become lower-dimensional). In contrast, topological similarity

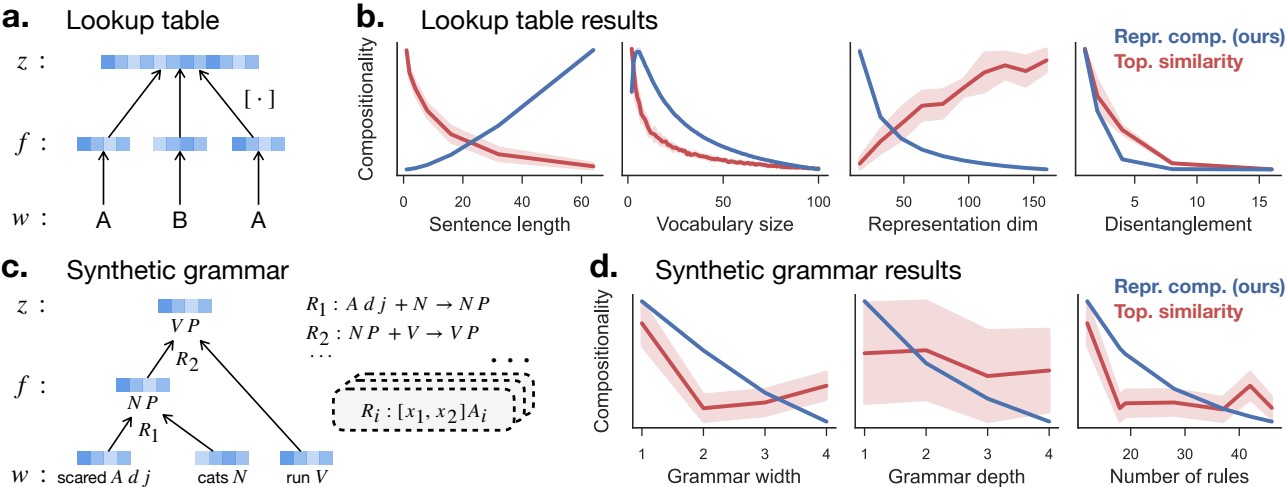

Figure 2. **Compositionality of synthetically-generated representations.** $C(Z)$ is consistent with intuitions about compositionality across all experiments, whereas topological similarity is not. **a.** In lookup table representations, words (or $n$-grams) are assigned embeddings which are concatenated to form $z$. **b.** Compositionality as a function of ground-truth representation properties. "Disentanglement" refers to varying $n$-gram size. **c.** In grammar representations, sentences are parsed with a context-free grammar, and each production rule is associated with a linear projection. Production rules are recursively applied, and the embedding at the parse tree's root defines $z$. **d.** Compositionality as a function of ground-truth properties of the grammar. Error bars show $\sigma$ over 10 seeds.

decreases with sentence length, thus violating intuitions.

*Vocabulary size:* If the vocabulary is too small relative to sentence length, then expressivity and compositionality are limited (e.g., with only one word, nothing can be expressed). On the other hand, if the vocabulary is too large relative to sentence length, then compositionality is low because expressivity doesn't come from combining constituent parts (e.g., with one-word sentences, there is no notion of parts). For a given sentence length, then, compositionality should peak at some intermediate vocabulary size. We observe this empirically with representational compositionality: a sharp early increase in compositionality followed by a monotonic decrease as vocabulary size increases further. In Appendix E, we confirm that this peak compositionality occurs at larger vocabulary sizes for larger sentence lengths. While topological similarity also decreases with vocabulary size, it does not show the early increase, and is in fact largest for a vocabulary size of 1.

*Representation dimensionality:* We increased representation dimensionality by increasing the dimensionality of the word embeddings. The representation grows more expressive with dimensionality, but from increased word complexity rather than word combinations. We should therefore expect compositionality to decrease. Representational compositionality empirically captures this phenomenon because the only thing increasing in this scenario is the size of the lookup table $K(f)$, so that $C(Z)$ decreases. Topological similarity, in contrast, increases as a function of representation dimensionality.

*Disentanglement:* When the meanings of words are context-dependent, a language is considered less compositional (e.g.,

idioms like "break a leg" are not considered compositional). Compositionality should therefore decrease as a function of disentanglement. Notably, a disentanglement of 1 in these experiments corresponds to a structure-preserving map, which has grounded some longstanding intuitions about compositionality (Montague et al., 1970). Representational compositionality empirically aligns with expectations by decreasing as a function of disentanglement because the size of the lookup table defining $K(f)$ grows exponentially. This supports the claims we made in Section 3 about how our definition encompasses prior intuitions around structure-preserving maps. Topological similarity follows the same trend as our definition in this case because it is designed to be maximal for a structure-preserving map.

**Context-free grammars** While our lookup table experiments provide intuitions for representational compositionality, they are unlikely to reflect the structure of representations in DNNs and brains. For instance, The Language of Thought hypothesis (Fodor, 1975) posits that representations underlying human thought have a hierarchical structure akin to context-free grammars in natural language (Chomsky, 1956). In such grammars, the meanings of sentences decompose according to parse trees, where children merge into parents through *production rules* and leaves correspond to words. For instance, the sentence "scared cats run" decomposes according to "ADJECTIVE (*scared*) + NOUN (*cats*) → NOUN-PHRASE (*scared cats*)" followed by "NOUN-PHRASE (*scared cats*) + VERB (*run*) → VERB-PHRASE (*scared cats run*)", where symbols such

as `NOUN-PHRASE` are *parts of speech* (similar to data types) and functions between parts of speech such as `NOUN+VERB → VERB-PHRASE` are *production rules*.

To model such systems using representational compositionality, we generated representations using simple synthetic grammars (Figure 2c). First, we assigned each word in a vocabulary an embedding and a part of speech, and we defined a grammar with a set of production rules. We then generated a dataset of sentences and parsed them using the grammar. Finally, the semantics were defined by embedding each word in the sentence and then applying a rule-specific function at every node in the parse tree until the root was reached, whose value we defined to be the representation. The rule-specific functions concatenated children embeddings and applied a linear projection.

We generated many synthetic representations in this way and measured their resulting representational compositionality (Figure 2d). For representational compositionality to match intuition, the number of rules in the grammar should be inversely proportional to compositionality. For example, in a natural language like English, we can express an infinite number of ideas using a relatively small set of grammatical rules and vocabulary, and this is why we believe natural language is compositional. We thus varied two properties of the grammar: its "width" and its "depth". Width refers to the number of rules that are defined for each level of the parse tree's hierarchy. Depth refers to the number of levels in the parse tree's hierarchy with unique rules prior to solely recursive application.

As both width and depth increase the complexity of the grammar, we should expect compositionality to decrease as a function of both. Representational compositionality is empirically consistent with this intuition because $K(f)$ increases as a function of the number of rules, each of which was associated with its own linear projection matrix. Topological similarity only loosely correlates with intuition, and has far more noise for different draws of $Z$ from the same grammar. Notably, these experiments support the theoretical claims made in Section 3 on how our definition relates compositionality to modularity, because each production rule serves as an independent module in $f$.

### 4.2. Emergent Languages from Multi-Agent Training

Next, we further validate our compositionality metric by applying it to real-world representations. To avoid having to solve the difficult optimization problem involved in measuring $C(Z)$ (which requires a minimization of $K(Z)$ w.r.t. $p_w$, $W$, $f$) we instead consider language systems in which $W = W^L$ is fixed and measure $C^L(Z)$ through Definition 3.

One interesting case of real language systems is those that emerge in multi-agent settings where agents must

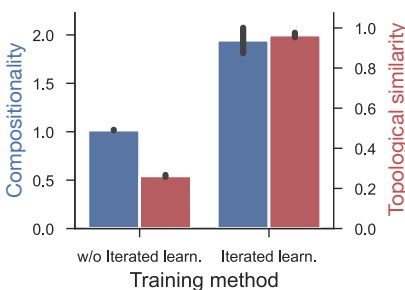

*Figure 3.* **Compositionality of language systems that emerge in multi-agent settings.** Our language system compositionality metric $C^L(Z)$ agrees with topological similarity on the ordering of models trained with and without iterated learning, but the numerical values provided by $C^L(Z)$ provide more theoretical insight (see main text). Error bars show $\sigma$ over 5 seeds.

learn to communicate. We consider the setting of Li & Bowling (2019); Ren et al. (2020) in which a speaker and a listener learn to communicate in a simple object reference game, where objects have symbolic attributes analogous to color, size, shape, etc. Agents trained using reinforcement learning typically communicate successfully, but often learn non-compositional language systems that arbitrarily map sentences to objects. However, Li & Bowling (2019); Ren et al. (2020) have shown that compositionality can emerge through a multi-generation process called *iterated learning* (Kirby et al., 2015), where the agents' parameters are periodically reset and pretrained on sentence/object pairs from the previous generation. Kirby et al. (2015) hypothesize that this occurs because iterated learning amplifies a model's inductive bias for simpler language systems that are more easily learnable across subsequent generations.

We trained agents both with and without iterated learning and measured $C^L(Z)$ for the resulting language systems. Training details are provided in Appendix J. After $N$ generations, we obtain a dataset consisting of all possible objects $Z$ and the sentences output by the speaker $W^L$ when given those objects as input. To measure $C^L(Z)$, we need both $K(Z)$ and $K(Z|W^L)$. Since $Z$ consists of a set of symbolic objects sampled uniformly, $K(Z)$ is simply equal to $|\mathcal{O}|\log_2(|\mathcal{O}|)$, where $\mathcal{O}$ is the set of all possible objects. To measure $K(Z|W^L)$, we used a compression method called prequential coding (Blier & Ollivier, 2018) because it provides good estimates in practice when DNNs are used for prediction (see Appendix H), but other compression techniques can be used if they provide tighter bounds. Prequential coding compresses $Z$ given $W$ by incrementally encoding individual datapoints $z_{<i}$ and fitting a model $\theta_{i-1}$ to predict them using $w_{<i}$ as input. The more datapoints are encoded, the better the model becomes by having seen more training data, and the more accurately it can predict the next datapoint $z_i$. Since prediction error is equivalent to complexity, $K(z_i|w_i, \theta_{i-1})$

will decrease as a function of $i$, which means that every subsequent datapoint takes fewer bits to encode. The total complexity $K(Z|W)$ is estimated by summing all of these terms.

Li & Bowling (2019) and Ren et al. (2020) measured compositionality using topological similarity. Using $C^L(Z)$, we find that we are able to reproduce their results (see Figure 3): iterated learning produces languages that are more compositional. However, a desirable property of our definition is that the absolute quantities of the metric are meaningful and interpretable. In particular, language systems trained without iterated learning obtain the lowest possible compositionality score, $C^L(Z) = K(Z)/K(Z|W^L) = 1$, meaning that the mapping from sentences to representations is entirely arbitrary. In contrast, topological similarity can at best only be used as a relative metric for comparing different languages, as its theoretical link to compositionality is not well understood.

### 4.3. Natural Languages

While it is commonly accepted that all natural languages are roughly equal in their expressive power (their ability to express ideas and thoughts), a highly debated question in linguistics is whether or not they are all equally compositional (Joseph & Newmeyer, 2012). This question has been difficult to answer definitively, partly due to the lack of principled and quantitative definitions of compositionality.

To investigate the compositionalities of natural language systems using our definition, we leveraged an existing dataset of English sentences describing natural images (COCO, 2024), which we then translated into French, Spanish, German, and Japanese using a large open source model (Costa-jussà et al., 2022). To obtain proxies of "meanings" $Z$ for these sentences, we encoded them using a multilingual LLM that embeds sentences to a dense fixed-size vector (Reimers & Gurevych, 2020). More experimental details as well as limitations of this approach can be found in Appendix K. Using these datasets of sentence/representation pairs, we measured the compositionalities of each natural language system $C^L(Z)$ using the same prequential coding approach as in Section 4.2. However, to estimate $C^L(Z)$ exactly, we would also have to estimate $K(Z)$ in the numerator for each language. While this quantity can be estimated in principle using compression techniques, for the sake of simplicity here we assume that these natural languages are all equally expressive in their abilities to express ideas and identify referents (i.e., equal $K(Z)$), which is a common assumption in linguistics. As a consequence, however, we cannot estimate the *absolute* compositionalities of these languages, but only their *relative* compositionalities; for this reason, compositionality in Figure 4 is measured in arbitrary units where we set the compositionality German to 1.

Our results are shown in Figure 4. Using prequential coding, we find that $K(Z|W^L)$ is similar for all languages, indicating that they have semantics $f^L$ of roughly equal

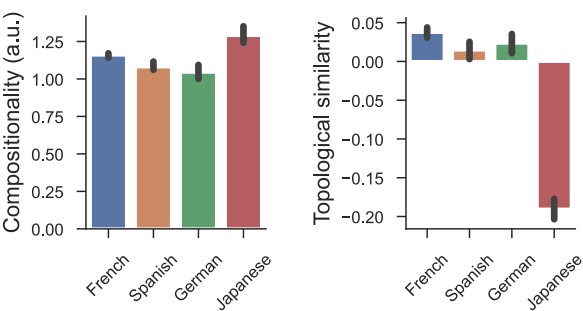

*Figure 4.* **Compositionality of natural language systems.** We consider language natural systems in which $W^L$ are sentences in some language and $Z$ are sentence embeddings obtained from a pretrained multilingual model. The relative compositionalities of languages measured using our definition $C^L(Z)$ are similar (left). Note that we do not estimate the numerator $K(Z)$ of $C^L(Z)$ and assume that it is constant across languages; as a result we can only compare *relative* compositionalities but do not know their absolute values (y-axis in arbitrary units, a.u.). **c.** Using topological similarity as a measure of compositionality gives counter-intuitive results (right): most languages have near-zero topological similarity and Japanese is a strong outlier with a topological similarity of $-0.2$. Error bars show $\sigma$ over 3 seeds.

complexity. This means that the relative compositionalities of these languages as measured by our definition $C^L(Z)$ are roughly equivalent, with Japanese being slightly more compositional. This result shows that our definition supports the *equal compositionality* side of the debate in linguistics surrounding natural languages that motivated this experiment at the start of the section (Joseph & Newmeyer, 2012). Using topological similarity as an alternative definition of compositionality gives counter-intuitive results that contradict our own: most languages have near-zero topological similarity, except for Japanese which is a strong outlier with a negative topological similarity of $-0.2$.

## 5. Related Work

**Communication and simplicity** Our definition is most closely related to experimental work in cognitive science which suggests that compositional languages evolve over time in human cultures due to competing pressures for effective communication and simplicity (Kirby, 1999; Kirby et al., 2004; 2008)—a result which has also been explored in AI (Li & Bowling, 2019; Ren et al., 2020; 2023; Ren & Sutherland, 2024) through emergent languages games described in Section 4.2. Our work builds on these foundations in several ways. First, we formalize the notions of a language (the tuple $(W, f, Z)$), the language's simplicity (simple semantics with low $K(f)$), and effective communication (expressive meanings through high $K(Z)$ with low information loss $K(Z|W, f)$). Second, Kirby's work is about the compositionality of a *language system* where $W$ is externally-defined

(our Definition 3), whereas we also provide a definition for representational compositionality where $W$ is intrinsically defined through optimal compression of $Z$. The latter notion is more directly relevant to the Language of Thought hypothesis (Fodor, 1975) and to the learned representations of DNNs.

**Representation reconstruction**   Many formal definitions frame compositional representations as those that can be reconstructed as a function of parts—our definition can be seen as an extension or generalization of many of these alternatives. First, reconstruction-based approaches to quantifying compositionality often measure the reconstruction error of a representation $Z$ from externally-defined parts $W$, where $W$ might be the input to the function that output $Z$ (Andreas, 2019; Ram et al., 2024), paired natural language data (Trager et al., 2023; Lewis et al., 2022), or underlying task latent variables (Wiedemer et al., 2024; 2023)—again representational compositionality extends these works by defining a $W$ that is intrinsic to $Z$ in a non-arbitrary way through optimal compression.

Second, prior work measuring compositionality through reconstruction from parts often makes strong assumptions about the *form* of the reconstruction for instance that it is a linear (Yun et al., 2021; Trager et al., 2023) or hierarchical (Andreas, 2019; Ram et al., 2024) function of word embeddings. In contrast, our definition makes no such assumptions and abstracts over arbitrary functions through the lens of their complexity $K(f)$, thereby generalizing prior work. When reconstruction-based approaches do not explicitly constrain complexity by imposing functional constraints (e.g., when they instead use use unconstrained DNN decoders), they neither minimize nor measure the complexity of the resulting semantics function, but instead measure the generalization ability of this function to novel sentences (Lewis et al., 2022; Bricken et al., 2023), which can be seen as a proxy for simplicity.

## 6. Limitations

Representational compositionality, while of theoretical interest for understanding compositionality, is uncomputable because its terms involve Kolmogorov complexities. Like Kolmogorov complexity, it can be estimated using tractable compression algorithms, but while we did this for $C^L(Z)$ we did not attempt it for $C(Z)$ in the current paper. We suggested one way that $C(Z)$ could potentially be estimated in Appendix B that involves training a discrete auto-encoder with a maximum likelihood prior in the latent space, but in practice this approach might be sensitive to modeling choices such as the DNN architectures, the training hyperparameters, and the quantity of training data. Indeed, related reconstruction-based approaches to measuring compositionality (Section 5) that impose tighter modeling constraints

such as linear semantics (Yun et al., 2021; Trager et al., 2023) might be inherently more stable than our approach, because there are fewer modeling choices to make. However, we also emphasize that the advantage of an abstract definition like ours is precisely that it allows practitioners to design a breadth of estimators that make different tradeoffs between flexibility and sensitivity to modeling choices. Empirical comparisons between our definition and other reconstruction-based measures is an important direction for future work. Finally, we defined compositionality with respect to discrete parts, but it may be possible to generalize this to continuous parts as well.

## 7. Conclusion

We introduced a novel definition of compositionality grounded in algorithmic information theory. Using theoretical arguments and empirical experiments, we showed that this simple definition not only accounts for our many intuitions about compositionality, but also extends them in useful ways.

In virtue of being quantitatively precise, representational compositionality can be used to investigate compositionality in real-world systems. We demonstrated this with emergent and natural languages where the sentences describing a representation are externally defined. In future work, this quantity can readily be applied to score tokenization schemes producing different representations in downstream models, which may lead to improvements in their design.

More generally, measuring the compositionalities of *representations* without a given mapping to sentences is an important direction for future work, as it will allow us to investigate the compositionalities of representations that emerge from different learning objectives, neural architectures, inductive biases, and brain regions. In turn, we will be able to see how representational compositionality empirically relates to other topics in ML such as compositional generalization, multitask generalization, and latent space generative models—we give some hypotheses and ideas for future work along these lines in Appendix F. In particular, representational compositionality has the potential to explain the success of varied methods because it defines compositionality through compression, which abstracts across the architecture, learning details, and particular representational format of a model. Representational compositionality can therefore be used to validate or reject diverse hypotheses about compositionality, such as the Language of Thought hypothesis (Fodor, 1975).

We hope that representational compositionality can also aid in the design of machine learning models with principled inductive biases for compositionality. Namely, in addition to supporting a given task, a compositional representation must be easily describable as a simple function of constituent parts; we describe some approaches for achieving this in Appendix G that can be pursued in future work.

## Acknowledgments

EE acknowledges support from Vanier Canada Graduate Scholarship #492702. YB acknowledges CIFAR, NSERC and Samsung for research funding. GL acknowledges support from a Canada-CIFAR AI Chair and the Canada Research Chair in Neural Computations and Interfacing (CIHR, tier 2) as well as from NSERC Discovery award RGPIN-2018-04821. GL is also a Visiting Researcher at Google's Paradigms of Intelligence Team.

## Impact Statement

This paper presents work whose goal is to advance the field of Machine Learning. There are many potential societal consequences of our work, none which we feel must be specifically highlighted here.

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

# A. Background on Kolmogorov complexity

Kolmogorov complexity was independently developed in the 1960s by Kolmogorov (1965), Solomonoff (1964), and Chaitin (1966), and defines a notion of "information quantity".

Intuitively, the Kolmogorov complexity of an object is the length of the shortest program (in some programming language) that outputs that object. Specifically, given some finite string $x$, $K(x)$ is the length $l(r)$ (in bits) of the shortest binary program $r$ that prints $x$ and halts. Let $U$ be a universal Turing machine that executes these programs. The Kolmogorov complexity of $x$ is then:

$$K(x) = \min_r \{l(r) : U(r) = x, r \in \{0,1\}^*\}, \tag{4}$$

where $\{0,1\}^*$ denotes the space of finite binary strings. A related notion is the conditional Kolmogorov complexity of a string $x$ given another string $y$, which is the length of the shortest program that takes $y$ as input and outputs $x$:

$$K(x|y) = \min_r \{l(r) : U(r(y)) = z, r \in \{0,1\}^*\}, \tag{5}$$

where $r(y)$ denotes a program taking $y$ as input. Finally, we can also define a "joint" Kolmogorov complexity $K(x,y)$, which denotes the length of the shortest program that jointly outputs both $x$ and $y$. Surprisingly, joint Kolmogorov complexity is related to conditional Kolmogorov complexity (up to an additive logarithmic term, which we will ignore) by the Symmetry of Information theorem (Li et al., 2008):

$$K(x,y) = K(y|x) + K(x) = K(x|y) + K(y). \tag{6}$$

Kolmogorov complexity has many intuitive properties that make it attractive as a measure of information quantity, and although it is less common than notions from Shannon information theory (Shannon, 2001), it is strictly more general (as we will show later below). The smaller and the more "structure" an object has—regularity, patterns, rules, etc.—the more it can be compressed using a short program and the lower its Kolmogorov complexity. Kolmogorov complexity therefore is deeply rooted in the idea of compression. For instance, a sequence with repeating patterns or a dataset that spans a low-dimensional subspace can be significantly compressed relative to its original size, and this results in low Kolmogorov complexity. In contrast, a random string devoid of any structure cannot be compressed at all and must in effect be "hard-coded", making its Kolmogorov complexity equal to its original size in bits.

While powerful, Kolmogorov complexity has certain limitations. First and foremost, Kolmogorov is intractable to compute exactly because it requires a brute force search over an exponentially large space of possible programs. It is therefore often of conceptual rather than practical value, although it can nevertheless be upper-bounded using more efficient compression strategies. Second, Kolmogorov complexity depends on the programming language of choice. For instance, if a programming language has a built-in primitive for the object being encoded, Kolmogorov complexity is trivially small. This concern, however, is often overblown: given any two Turing-complete programming languages, the difference in Kolmogorov complexity that they assign to an object is upper-bounded by a constant that is independent of the object itself, because any Turing-complete programming language can simulate another (Grünwald & Vitányi, 2003; Fortnow, 2000). In practice, we can simply consider "reasonable" Turing-complete programming languages that don't contain arbitrary object-specific primitives, in which case this simulation constant will be relatively small and the particular programming language of choice will have little effect. Finally, Kolmogorov complexity is only defined for discrete objects because no terminating program can output a continuous number with infinite precision. This concern is also less consequential in practice, because we can always represent continuous objects using finite (e.g., floating-point) precision.

**Important properties for machine learning**  In ML, we are often concerned with datasets and probabilistic models. Kolmogorov complexity relates to these two concepts in several interesting ways. First, we can ask about the Kolmogorov complexity of a finite dataset $X = (x_1,...,x_n)$ where each sample is drawn *iid* from a distribution $p(x)$. It turns out that if we have access to the true distribution $p(x)$, optimal algorithms such as arithmetic coding (Witten et al., 1987) can encode each sample using only $\log_2 p(x_i)$ bits. Intuitively, this is because samples that occur more frequently can be encoded using shorter codes in order to achieve an overall better compression. We thus have that:

$$K(X|p) = -\sum_{i=1}^{n} \log_2 p(x_i). \tag{7}$$

If instead of access to the true distribution $p(x)$ we only have a probabilistic model of the data $p_\theta(x)$, we have that:

$$K(X|p_\theta) \leq -\sum_{i=1}^{n} \log_2 p_\theta(x_i), \tag{8}$$

where we have equality when $p_\theta = p$. This insight is significant. Notice that $-\sum_{i=1}^{n} \log_2 p_\theta(x_i)$ is the negative log-likelihood of the data under the model, which is a common loss function used in ML. This tells us that models with lower error better compress their data, and directly relates Kolmogorov complexity to optimization in ML. However, what if we do not have a model? What is the Kolmogorov complexity of the data itself? Intuitively, if the dataset is sufficiently large, the optimal method for encoding it should be to first specify a model and then encode the data using that model as in Equation (8). Specifically, using identities in Fortnow (2000), we have:

$$K(X) \leq K(X|p_\theta) + K(p_\theta). \tag{9}$$

This encoding scheme on the RHS is referred to as a 2-part code (Grünwald, 2007). We have equality when the model's description length and error are jointly minimized, which occurs when the model $p_\theta(x)$ is equivalent to the true distribution $p(x)$:

$$K(X) = \underset{p_\theta}{\arg\min} K(X|p_\theta) + K(p_\theta) = \underset{p_\theta}{\arg\min} -\sum_{i=1}^{n} \log_2 p_\theta(x_i) + K(p_\theta) \tag{10}$$

$$= K(X|p) + K(p) = -\sum_{i=1}^{n} \log_2 p(x_i) + K(p). \tag{11}$$

Again, we can draw important connections to ML. Equation (9) says that the Kolmogorov complexity of a dataset is upper-bounded by the a model's error and complexity. In addition, Equations (10) and (11) tell us that the simplest model that explains the data is most likely to be the true one, which draws a theoretical link between compression, maximum likelihood training, model complexity, and generalization (Goldblum et al., 2023).

**Relation to Shannon information**   In Shannon information theory (Shannon, 2001), the notion of information quantity is entropy. Given a random variable $X \sim p(x)$, entropy is defined as: $H(X) = \mathbb{E}_{x \sim p(x)} -\log_2(p(x))$. Notice that the $-\log_2(p(x))$ inside the expectation is equal the quantity inside the sum of Equation (7), which specified the minimum number of bits needed to encode a sample from a dataset given the distribution that sample was drawn from. This is no accident: entropy can be seen as the average number of bits needed to compress events from a distribution using an optimal encoding scheme when the distribution $p(x)$ is known. If we simply sum these bits for a finite number of samples instead of taking an expectation, we get exactly $K(X|p)$ as defined in Equation (7).

As we have seen, though, the assumption about a known distribution $p(x)$, need not be made in the Kolmogorov complexity framework. In this sense, Kolmogorov complexity is a strict generalization of Shannon information theory: $K(X)$ as defined in Equation (11) is equivalent to summed entropy plus the complexity of the distribution $p(x)$, which is unknown and needs to be encoded. In the Shannon framework, it is difficult to derive a meaningful notion for the information quantity in the distribution $p(x)$ because it is an individual object—a function, in particular—and Shannon information is only defined for random variables (Grünwald & Vitányi, 2003). A second drawback of Shannon information is that entropy is a measure of statistical determinability of states; information is fully determined by the probability distribution on states and unrelated to the representation, structure, or content of the individual states themselves (Grünwald & Vitányi, 2003). For this current work, we require a notion of complexity that can account for representations and functions, making Kolmogorov complexity better suited to the task.

## B. Compressing a representation using discrete auto-encoders

To measure compositionality as defined in Definition 2, we must first compress $K(Z)$ using the program form in Section 2. This involves finding a $p_w, W$, and $f$ that jointly minimize:

$$K(Z) = \min_{p_w, W, f} K(p_w) + K(W|p_w) + K(f) + K(Z|W, f) \tag{1 revisited}$$

$$= \min_{p_w, W, f} K(p_w) - \sum_{n=1}^{N} \log p_w(w_n) + K(f) - \sum_{n=1}^{N} \log \mathcal{N}(z_n; f(w_n)).$$

While this is an intractable search problem, it can be turned into an easier optimization problem using modern deep learning tools. In particular, we can minimize at least some of the terms in Equation (1) by fitting a discrete auto-encoder to $Z$ using a learned prior in the latent $W$-space, as illustrated in Figure B.1. This auto-encoder consists of an encoder $w = e(z)$ that maps the representation to a discrete latent space of sentences, a latent prior $p_w(w)$, and a decoder $\mathcal{N}(z; f(w))$ that outputs the sufficient statistics of a Gaussian distribution in order to evaluate the likelihood of the original representation. In practice, the latent prior $p_w(w)$ can be parameterized using an auto-regressive model such as a causal Transformer, which tends to work well on language data. We can then train this discrete auto-encoder using the following loss function:

$$\mathcal{L}(Z; e, p_w, f) = \sum_{z \in Z} -\log p_w(e(z)) - \log \mathcal{N}(z; f(e(z))). \tag{12}$$

The first term in this loss ensures that $W$ has high prior likelihood, and optimizes both the prior model $p_w$ as well as the encoder $e$ that produces the latent sentences. The second term in the loss ensures that $Z$ has high likelihood given $W$, and optimizes the decoder $f$ as well as the encoder $e$ so that they preserve information about $Z$. Recall from Equation (7) that the negative likelihood of an object under some probability distribution is equal to its conditional Kolmogorov complexity given that distribution. As a result, minimizing the loss in Equation (12) is equivalent to finding a $p_w$, $W$, and $f$ that jointly minimize $K(W|p_w) + K(Z|W, f)$.

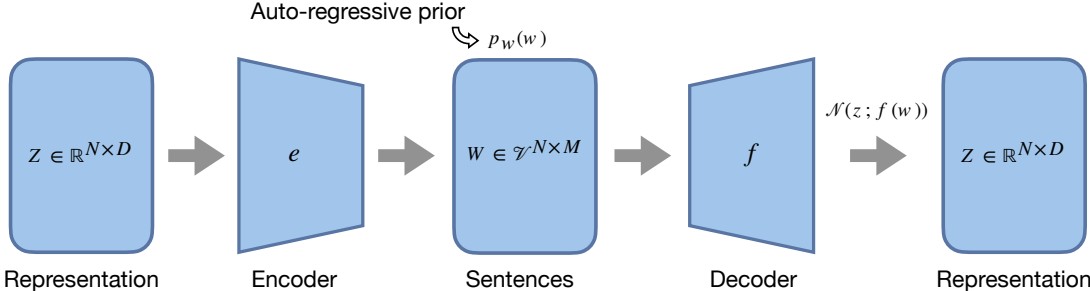

1. Fit a discrete auto-encoder with learned prior

$$\mathcal{L} = -\log p_w(W) - \log p(Z \,|\, f(W))$$

2. Measure complexity terms

$$K(Z) = K(p_w) + K(W \,|\, p_w) + K(f) + K(Z \,|\, W, f)$$

*Figure B.1.* **Estimating the complexity of a representation** $K(Z)$ **by fitting a discrete auto-encoder with learned latent prior.** The encoder, prior, and decoder are jointly trained with a loss that maximizes the likelihood of $Z$ using sentences that have high prior likelihood $p_w(W)$. If $p_w$ and $f$ are also regularized to be simple functions, fitting this discrete auto-encoder is equivalent to finding a $p_w$, $W$, and $f$ that jointly minimize $K(Z)$.

To measure $K(Z)$, we also need to minimize $K(p_w)$ and $K(f)$. For this, two options present themselves:

1. Hope that the implicit simplicity bias of DNNs trained using SGD does a good enough job on its own of finding solutions with low complexity (Blier & Ollivier, 2018).

2. Use additional regularization techniques that implicitly minimize the complexities of the models, such as simple architectures, L1 or L2 weight penalties, modularity (Goyal & Bengio, 2022), dropout (Hinton et al., 2012), periodic resetting (Zhou et al., 2021), etc.

Regardless of which method is used, the complexities of the final trained models can be estimated using a method called prequential coding (Blier & Ollivier, 2018), which we describe in Appendix H. Thus, we are able to estimate all of the constituent complexity terms of $K(Z)$ in Equation (1). The main challenge in this overall approach then becomes how to successfully train a discrete auto-encoder with a prior in latent space, in a way that is both stable and scalable.

**VQ-VAE** The most popular method for training discrete auto-encoders is the Vector-Quantized Variational Auto-Encoder (VQ-VAE) (Van Den Oord et al., 2017). While the latent prior in a VQ-VAE is generally trained post-hoc, some work has managed to train the prior end-to-end along with the rest of the model (Jones & Moore, 2020; Yasuda et al., 2021; Cohen

et al., 2022). The main challenge with VQ-VAEs is that they explicitly discretize in the latent space during training—which is an inherently non-differentiable operation—and then attempt to approximate gradients using imperfect estimators (Bengio et al., 2013; Jang et al., 2016). As a result, training is often unstable and fraught with degenerate solutions that collapse in the latent space (Łańcucki et al., 2020).

**Simplicial embeddings**    Another option, which avoids the difficulty of training with hard-discretization, is to use so-called *simplicial embeddings* in the latent space (Lavoie et al., 2023). Simplicial embeddings amount to soft attention: each vector "chunk" representing a word in the latent space is projected onto $|\mathcal{V}|$ word embeddings followed by a softmax, and the weighted word embeddings are then summed at each sentence position. The temperature of the softmax can then be gradually decreased over the course of training such that the operation approaches a hard-discretization in the limit. As the operation is entirely continuous and deterministic, it is easier to train using end-to-end gradient descent methods (although it may become numerically unstable at low softmax temperatures). One challenge becomes how to define and train the prior $p_w$ in this case, where $W$ is in fact a sequence of continuous word embedding mixtures as opposed to a sequence of discrete tokens. One possibility is to perform a hard-discretization of the latent before it is passed to the prior, along with relevant gradient estimators (e.g. Bengio et al., 2013; Jang et al., 2016). While this could make training more difficult, the encoder-decoder part of the model would at least remain entirely continuous and deterministic. Another option is to define $p_w$ in continuous space, where the input is a sequence of word embedding mixtures and the "next-token" targets are categorical distributions over words.

**GFlowNets**    If we still wish to perform hard-discretization, but do not want to resort to imperfect gradient estimators required for end-to-end training, Generative Flow Networks (GFlowNets) could be a promising alternative (Bengio et al., 2021; 2023). GFlowNets can learn to sample some compositional discrete object in proportion to a reward function. The reward function and GFlowNet can also be conditioned on some input, and the reward function can be learned in alternation with the GFlowNet using expectation-maximization (GFlowNet-EM) (Hu et al., 2023). In the case of a discrete auto-encoder, the encoder would be a GFlowNet, while the decoder and prior would be the reward function. While this approach has been used to train a discrete auto-encoder before (Hu et al., 2023), it comes with its own challenges. First, GFlowNet-EM is not an end-to-end training procedure (no gradients flow from the decoder to the encoder), which makes it more difficult to train. Second, while GFlowNets sample proportionally to their reward, our ultimate goal is to *maximize* the reward (i.e., find sentences $W$ that maximize the prior and reconstruction). To do this, we will ultimately have to decay the temperature of the reward over the course of training in order to settle to a final solution that minimizes the loss in Equation (12). Training GFlowNets with a sparse reward, however, is more difficult due to exploration challenges (Atanackovic & Bengio, 2024).

**Computational complexity**    If the discrete auto-encoder described in this section can be trained successfully, then estimating representational compositionality is tractable, despite being defined theoretically in terms of Kolmogorov complexities. Fitting the auto-encoder itself is tractable using modern machine learning hardware. Then, to estimate $K(p_w)$ and $K(f)$ we must use prequential coding (see Appendix H), which amounts to fitting a neural network at varying dataset sizes. While fitting a neural network $N$ times (where $N$ is the dataset size) is inefficient, it is nonetheless tractable, and can be approximated efficiently by chunking the data into coarser sizes as we did in our experiments. There are also methods for computing prequential coding online rather than retraining the model from scratch each iteration (Bornschein et al., 2022).

## C. Assumptions in compressing a representation

In laying out our framework for measuring $K(Z)$ in Section 2, we made several key assumptions.

First, we assumed that the shortest program that outputs $Z$ has a particular form. If it does not, then the estimated $K(Z)$ can be far greater than the true one. However, we argue that the assumed program form is safe for the kinds of representations that we are interested in and the kinds of insights we wish to gain from estimating $K(Z)$. Namely, we are interested in seeing if given neural representations share similar properties to conscious human thought, which is believed to have a symbolic structure where each thought is a composition of discrete concepts (Fodor, 1975). If a representation does not have this kind of structure, then our method would detect it in the form of a high estimated $K(Z)$, even if this is an overestimate of the true Kolmogorov complexity due to incorrectly assuming the program form in Section 2.

Second, actually estimating $K(Z)$ using Equation (1) requires a minimization over $p_w$, $W$, and $f$. This optimization approach assumes that the $p_w$ and $f$ which minimize $K(Z)$ are DNNs. While this can seem unintuitive at first given the significant number of parameters in DNNs, it has been found that they converge to solutions that are remarkably simple and compressible (Blier & Ollivier, 2018; Goldblum et al., 2023; Sutskever, 2023; Rae, 2023), which likely explains their strong generalization

abilities. We therefore believe that for neural representations with sufficient complexity, the assumption that they can be best compressed using DNNs is justified.

## D. Examples of compositional representations

To supplement and clarify the arguments in Section 3, it is easiest to gain further intuition for our definition of compositionality through concrete examples of different hypothetical representations. For each, we have strong intuitions about whether or not the representation is compositional, and we will see that our definition agrees with—and indeed extends—these intuitions. We illustrate these examples in Figure D.1.

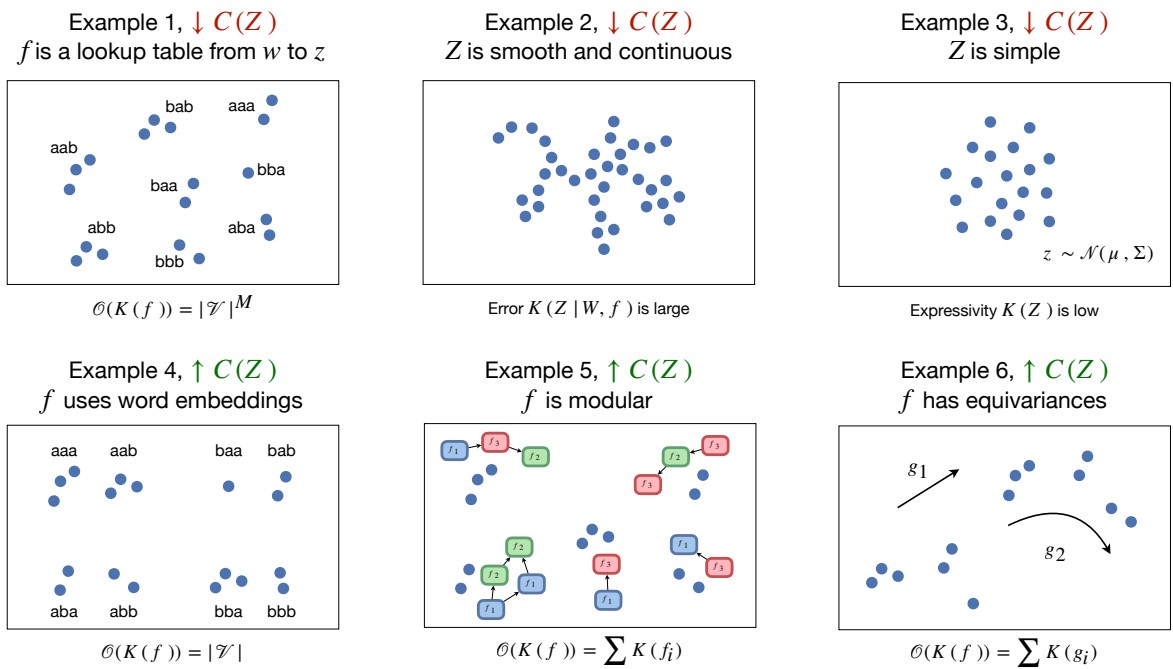

*Figure D.1.* **Examples of different representations and their compositionalities according to** $C(Z)$**. Example 1.** A representation whose clusters lack any structure has semantics $f$ that map $w \to z$ arbitrarily using a lookup table, resulting in high $K(f)$ and low $C(Z)$. **Example 2.** A representation that is smooth and continuous cannot be compressed as a function of discrete parts without incurring significant prediction error, resulting in high $K(Z|W,f)$ and low $C(Z)$. **Example 3.** A representation that cannot express many different things (thoughts, visual scenes, ideas, etc.), such as one that is sampled from a unimodal distribution, has low $K(Z)$ and low $C(Z)$. **Example 4.** A representation that can be described by assigning word embeddings which are then processed using a simple operation (e.g., concatenation, as in disentanglement) has low $K(f)$ and high $C(Z)$. **Example 5.** A representation whose semantics can be compressed using a small number of simple and reusable modules has low $K(f)$ and high $C(Z)$. **Example 6.** A representation whose semantics have a large number of symmetries, or equivariances, has low $K(f)$ and high $C(Z)$.

**Example 1,** $\downarrow C(Z)$**:** $f$ **is a lookup table from** $w$ **to** $z$    Consider a representation $Z$ that is sampled from a mixture of Gaussians, where the centroids are far apart but their locations lack any kind of structure (i.e., they are randomly distributed). To simplify things, let us assume that there are as many unique centroids as there are possible sentences. In such a case, the semantics function $f$ would identify each centroid with a unique sentence and the resulting error $K(Z|W,f)$ would be low. However, because these centroids lack any structure, $f$ would have to define an *arbitrary* mapping from each sentence to its corresponding centroid. In other words, $f$ would function as a lookup table from $w$ to $z$ that does not leverage the internal structure (i.e., words and their ordering) in the sentence to achieve a more compressed mapping. The resulting description length of $f$ would be equal to the size of the lookup table, which would grow exponentially with the sentence size. $f$ would be, in effect, a complex "hard-coded" mapping (in fact, the most complex possible) with $\mathcal{O}(K(f)) = |\mathcal{V}|^M$, where $M$ is the sentence length and $|\mathcal{V}|$ is the vocabulary size. The resulting compositionality $C(Z)$ would be extremely low.

**Example 2,** $\downarrow C(Z)$**:** $Z$ **is smooth and continuous**    The above example considered a case where the representation had discrete structure that could be accurately modeled by sentences, and the source of low compositionality came from a high

$K(f)$. However, the compositionality can also be low if $Z$ is inherently continuous, in which case modeling it using a discrete $W$ is at best an approximation via quantization. In such a case, the error $K(Z|W,f)$ would be high and the corresponding compositionality would be low. Note that it might be possible to compress $Z$ using a low-dimensional continuous code rather than discrete sentences, from which an equivalent (perhaps even identical) definition of continuous compositionality could be derived, but in this work we consider only compositions of discrete parts.

**Example 3, $\downarrow C(Z)$: $Z$ is simple**     Most of the discussion thus far has focused on the denominator of $C(Z)$ in Definition 2. However, a representation can also lack compositionality if the complexity of the numerator, $K(Z)$, is low. If $Z$ were very low—say it were a constant, for instance—then it could be modeled using a simple $f$ that achieves low error $K(Z|W,f)$. However, we would certainly not be tempted say that the representation is compositional. In fact, it would be best compressed using a single word and an $f$ that outputs a constant, rather than using complex sentences and simple compositional rules. Compositionality must therefore also increase with the expressivity of the representation, which is captured by the numerator $K(Z)$ in our definition. In cognitive science, where the scientific notion of compositionality has its origins, expressivity is considered an essential component of compositionality; Chomsky (1956) famously argued that natural language as a compositional system derives its power because it gives us "infinite use of finite means", or in the language of our definition high expressivity as a simple function of parts.

**Example 4, $\uparrow C(Z)$: $f$ assigns an embedding to each word followed by a simple operation**     We now turn to paradigmatic examples of high compositionality, beginning with the most intuitive. Consider once again a representation $Z$ that is sampled from a mixture of Gaussians like in *Example 1*, but this time imagine that the centroids are arranged in a structured way. In particular, imagine that they are structured such that each can be described as a concatenation of subcomponents that are shared across all centroids. Now, the simplest $f$ would be one that first assigns a vector embedding to each word such that it represents a possible subcomponent of the centroid, and then concatenates the embeddings for all words in the sentence. The complexity of $f$ would then scale only linearly as a function of the number of words in the vocabulary (assuming they are all necessary), because concatenation is a simple operation that takes a constant number of lines of code. We would have $\mathcal{O}(K(f)) = |\mathcal{V}|$, which is independent of the sentence length, in contrast to the arbitrary mapping in *Example 1* that scaled as $\mathcal{O}(K(f)) = |\mathcal{V}|^M$. This is a substantial reduction in complexity and increase in compositionality, and it comes from the fact that the words contribute independently to the representation. This is a case of a perfectly disentangled representation, which in our theory is simply an extreme case of compositionality, but intermediate cases are possible as well. For instance, the representation could be determined by interactions between pairs of words in the sentence, or it might be the case that words largely contribute independently to the representation but that there is some small degree of context-sensitivity, as in human language. Our theory unifies all of these cases under a single, succinct definition.

**Example 5, $\uparrow C(Z)$: $f$ is modular**     As already explained in Section 2, a modular $f$ is simpler to describe and thus implies higher compositionality. To see why modular functions are more compressible, consider a paradigmatic case: computer programs. When a computer program is written in such a way that it can be refactored into a small number of functions and classes that are reused several times, the total length of the program decreases substantially. Programs that are not written with modularity in mind tend to be much longer and complex. Modular functions therefore tend to have far lower complexity because the modules only need to be defined once, but can then be reused many times inside the function. In ML, modularity is leveraged in a similar fashion. For instance, Goyal et al. (2021) introduces an architecture that consists of $N$ DNNs as well as a learned attention-based routing mechanism for how they communicate. Crucially, these modules are leveraged by the routing mechanism in a context-dependent way, and each module can be reused many times to process each individual input. This means that while the entire model is simple (small number of modules and simple routing mechanism), it is nevertheless highly expressive due to the combinatorial way in which modules can be composed. Our definition explains how this expressivity and compression endowed by modular functions formally relates to compositionality (Lepori et al., 2023; Goyal & Bengio, 2022).

**Example 6, $\uparrow C(Z)$: $f$ has many equivariances**     The connection between equivariance and compositionality is perhaps less obvious (Gordon et al., 2020), but it is a natural and intuitive consequence of our definition. Equivariances (and invariances) are symmetries—sources of structure that decreases the complexity of a function (Immer et al., 2022; Wilk et al., 2018; van der Ouderaa & van der Wilk, 2022). For instance, convolutional layers have local connectivity and reuse weights across spatial locations, which both reduces their description length and makes them equivariant to spatial translations. We can also consider linear equivariance as a special case that is easy to illustrate. If $f$ is linearly equivariant to a particular operation $g$ in sentence-space, it means that $f(g(w)) = f(w) + v_g$, where $v_g$ is a constant vector that corresponds to the equivariant change in the representation output by $f$. The difference in the function's behaviour for two different inputs, $w$ and $g(w)$, can therefore be compactly de-

scribed by a single vector, whereas in the general non-equivariant case the change in the function's behaviour can be arbitrarily complex. In an extreme case, if $f$ can be completely described by a set of linear equivariances, then each $w$ corresponds to a set of $g_i$'s applied to a constant "default" sentence, and $f$ merely needs to encode a single vector for each of these $g_i$'s then sum those that apply to a particular input. The resulting function is very similar to the one described in *Example 4*, where $f$ applied a simple operation to a sequence of word embeddings in a sentence (in this case vector addition). The function also bears similarities to the one described in *Example 5* if we view the equivariances as modules. Similar arguments can be made for non-linear equivariance, where the complexity $K(f)$ would still be reduced, but to a lesser extent. In general, the more equivariances a function has and the simpler those equivariances are, the lower the complexity $K(f)$ and the higher the compositionality $C(Z)$.

## E. Synthetic representations: varying both sentence length and vocabulary size in lookup tables

In Section 4.1 when discussing our results on synthetic representations generated from lookup tables, we argued that sentence length and vocabulary size should have a *joint* impact on compositionality. Namely, if the vocabulary is too small relative to sentence length, then expressivity and compositionality are limited (e.g., with only one word in the vocabulary, nothing can be expressed). On the other hand, if the vocabulary is too large relative to sentence length, then compositionality is low because expressivity doesn't come from combining constituent parts (e.g., with one-word sentences and a large vocabulary, there is no notion of parts). A good quantitative definition of compositionality should capture this intuition. In Figure E.1, we demonstrate that our definition of representational compositionality reproduces this result empirically: as sentence length grows, the vocabulary size that maximizes compositionality does as well.

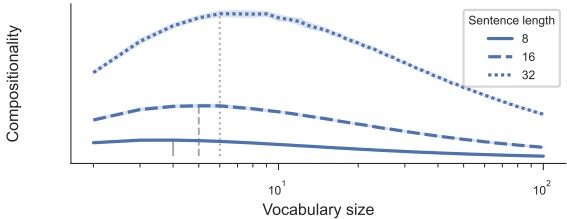

*Figure E.1.* **Compositionality** $C(Z)$ **for synthetically-generated lookup table representations as a function of both vocabulary size and sentence length.** As in Section 4.1, lookup table representations are generated by uniformly sampling sentences of a particular length and vocabulary size, and mapping individual words to vectors in a lookup table, followed by concatenation. As sentence length grows, the vocabulary size that maximizes compositionality does as well. Note that for clarity, vocabulary size is shown on a log scale. Vertical lines show peaks. Error bars show $\sigma$ over 10 seeds.

## F. Relations between representational compositionality and other ML topics

**Compositional generalization**    One of the benefits of compositional representations is that they enable better *compositional generalization* (Lake & Baroni, 2018). If a model is compositional with respect to a set of features in its training data, it need not observe all possible combinations of those features in order to generalize to novel ones (Schug et al., 2024; Wiedemer et al., 2024; 2023; Bahdanau et al., 2019; Mittal et al., 2021; Hupkes et al., 2020; Jarvis et al., 2024; Lippl & Stachenfeld, 2024; Lachapelle et al., 2024). For instance, if an image classifier's representation is compositional with respect to foreground objects and background scenes, then it should be able to correctly classify an image of "a cow on a beach" at inference time after having only observed cows and beaches separately at training time.

In certain cases, compositionality is defined in terms of a model's ability to compositionally generalize compositionally (e.g., Jarvis et al., 2024; Wiedemer et al., 2024; 2023; Lippl & Stachenfeld, 2024). However, while such definitions of compositionality can often provide theoretical guarantees on generalization, they also place strong assumptions on either the representation, the downstream model, or both. For instance, Wiedemer et al. (2023) assumes that the representation is perfectly disentanglement with respect to some underlying task constituents. Similarly, Lachapelle et al. (2024) assumes disentanglement and that the downstream function using the representation is additive with respect to the the disentangled factors, and Lippl & Stachenfeld (2024) assumes disentanglement and "conjunction-wise additivity". Wiedemer et al. (2024) takes from the object-centric learning literature and defines a compositional representation as one that is structured into distinct "slots" (Locatello et al., 2020), and then requires that the downstream model using these slots is additive.

In contrast, our definition of representational compositionality is far more general: it defines compositionality in terms

of compression, which abstracts across the architecture producing and using the representation, learning details, data requirements, and particular representational format. For instance, disentangled and slot-wise representations are particular cases of representational compositionality in terms of their simple semantics $K(f)$ (see Appendix D), but these are rigid assumptions to build into a model that might negatively impact performance. In contrast, representational compositionality has the potential to explain the success of more varied and flexible methods in terms of compositional generalization, such as loss regularizers or simply scaling dataset and model size.

As a consequence of its generality, it may be difficult to formally characterize the relationship between representational compositionality and compositional generalization with theoretical guarantees, and we did not attempt to do so in this paper. Nevertheless we hypothesize that representations with high $C(Z)$ should enable better compositional generalization. This is because the representation of constituent parts is systematic: the semantics mapping constituent parts to the representation is a simple function that will generalize better to novel part combinations (i.e., it will assign them a meaningful rather than arbitrary representation, which downstream functions should be able to leverage). One of our central goals for future work is to test this hypothesis empirically, where we measure the compositionalities of many model representations using our definition and then correlate this score with the models' compositional generalization abilities.

**Generative models in latent space**    In addition to compositional generalization, representational compositionality also relates to generative models that sample in latent space. In particular, once a compositional representation is learned, efficient and generalizable generative models can be constructed by sampling in the space of discrete sentences, rather than in the high-dimensional continuous latent space directly. This is because the semantics function $f$ of a representation with high $C(Z)$ is simple, and can generalize to novel sentences that the generative model might produce. Empirically, modeling and sampling from discrete distributions is often easier and more effective, especially for complex multi-model distributions (Razavi et al., 2019).

To give a concrete example, imagine that a vision model has been pretrained on some task like object classification and produces latent representations with high $C(Z)$. Using this representation, we can train a generative model of the form $z \sim p_w(w)\mathcal{N}(z; f(w))$ described in Section 2, and then generate novel samples for downstream visual reasoning tasks directly in the abstract latent space, rather than in the low-level image space. This is similar to thought and reasoning involved in human cognition, which are generative processes believed to exhibit a discrete language-like structure (Fodor, 1975; Dehaene et al., 2022; Lake et al., 2017; Bengio, 2017; Goyal & Bengio, 2022).

## G. Inductive biases for representational compositionality

In virtue of being formally precise and quantitative, representational compositionality can inspire the design of novel inductive biases for compositional representations in ML models. In this section, we outline two approaches that we believe have promise: one that directly optimizes for $C(Z)$, and another that indirectly attempts to increase it through task and data constraints. In addition, $C(Z)$ can be used to validate existing inductive biases for compositionality (e.g., architectures for object-centric representations Locatello et al., 2020).

**Regularizing $K(Z|W)$**    The most direct way to learn representations with high $C(Z)$ is to regularize the denominator $K(Z|W)$ so that the representations become more *verbalizable*, as suggested in Bengio (2017) and Goyal & Bengio (2022). Definition 2 says that compositional representations are (a) expressive and (b) easily described using sequences of discrete symbols—in other words, that they are verbalizable like human thoughts that can largely be conveyed in natural language. Expressivity can be obtained simply by training on a sufficiently complex task; for example, representations for image classification need to be expressive so that they can discriminate different objects. Task pressure alone, however, does not guarantee that the representation will be verbalizable. This second desiderata can be achieved, however, through a prior that regularizes the model's loss function.

Say that some model $g_\theta$ produces a representation $Z = g_\theta(X)$ of inputs $X$. Verbalization corresponds to minimizing the denominator in Definition 2: $K(Z|W) = K(f) + K(Z|W, f)$. Crucially, $W$ and $f$ here are obtained from the shortest program that outputs $Z$ as described in Section 2, which can be approximated by optimizing a discrete auto-encoder who's training scheme is sketched out in Appendix B. To make the dependence of $W$ and $f$ on $Z$ more explicit here, we will use the superscripts $W^Z$ and $f^Z$. If we wish to increase verbalizability (and therefore compositionality), we therefore need to perform some update update $\theta \to \theta'$ such that:

$$K(f^{Z'}) + K(Z'|W^{Z'}, f^{Z'}) < K(f^Z) + K(Z|W^Z, f^Z), \tag{13}$$

where $Z' = g_{\theta'}(X)$. One option for accomplishing this is by backpropagating the reconstruction error of the discrete auto-encoder, $K(Z|W^Z, f^Z)$. This approach assumes that the semantics before and after the update are unchanged (i.e., $f^{Z'} = f^Z$), so that the only thing that needs to be considered is the auto-encoding reconstruction error $K(Z|W^Z, f^Z) \to K(Z'|W^{Z'}, f^{Z'})$. While this assumption will be violated in practice, it may hold approximately such that regularizing reconstruction error alone is sufficient to increase compositionality.

In sum, the approach described here consists of training a DNN $g_\theta(X)$ on some task as usual, but with an additional loss: a discrete auto-encoder is fit to a layer in the model which we want to be more compositional, and the $\theta$ is regularized to minimize the loss of this discrete auto-encoder. As a result, in addition to subserving task demands, the representation is optimized to be more compressible as a function of constituent discrete parts (i.e., it is verbalizable).

**Multi-task training**    A common observation in deep learning is that the model representations after training tend to be surprisingly simple despite the significant number of parameters in the network (Blier & Ollivier, 2018), as evidenced by their strong *iid* generalization abilities. However, absent additional constraints (e.g., Lachapelle et al., 2024), these same representations do not enable compositional out-of-distribution generalization, suggesting that they lack sufficient compositional structure. One hypothesis is that *while the simplest representation used to solve a single task may not be compositional, the simplest representation used to solve many related tasks might be*. An analogy can be made to computer programs. When a program is written for a single narrow purpose, writing it in a compositional manner that reuses shared functions and classes might in fact result in bloat that increases the total program length. However, if these same functions and classes constitute a useful library that can be leveraged to write other programs as well, significant compression might be possible because the library is shared across all programs.

In the terminology of $C(Z)$, learning the simplest representation that subserves many different related tasks might result in low $K(Z|W)$ and high compositionality because the semantics $f$ are shared across these tasks and therefore lead to high compression; only $K(p_w)$ grows to accommodate additional tasks, analogous to how a programming library would be used in novel ways to write a new program. Since DNNs already tend to learn simple representations (Blier & Ollivier, 2018), our definition suggests that ordinary training in certain multi-task settings (those that reuse certain task components) might be a simple method for learning compositional representations. Indeed, this has long been hypothesized and observed empirically (Driscoll et al., 2024; Johnston & Fusi, 2023; Lachapelle et al., 2023; Vafidis et al., 2024a; Maziarka et al., 2022; Vafidis et al., 2024b), especially in the case of disentangled representation learning, and could be verified more formally using our definition of representational compositionality.

## H. Prequential coding

While the Kolmogorov complexity of a model $K(p_\theta)$ is difficult to measure directly, it turns out that we can jointly estimate $K(D|p_\theta) + K(p_\theta)$ in cases where the model was fit to the data using a learning algorithm, as is the case in ML. From Equation (6), we have that:

$$K(D|p_\theta) + K(p_\theta) = K(D, p_\theta). \tag{14}$$

Instead of trying to estimate the terms on the LHS directly, we can estimate the RHS by finding the shortest program that jointly compresses both the dataset and the model, which we turns out to be easier through a compression algorithm called *prequential coding* illustrated in Figure H.1 and described below.

Prequential coding first assumes that we have access to a learning algorithm $T$ which was used to fit the model $p_\theta$. For instance, $p_\theta = T(D)$ might correspond to a randomly initialized DNN architecture fit to $D$ using SGD with some set of hyperparameters. Then, consider an ordering of *iid* datapoints $D = \{D_1, ..., D_N\}$, and denote $D_{1:i} = \{D_1, ..., D_i\}$. In prequential coding, the first datapoint $D_1$ is hard-coded in an uncompressed form, which takes a large number of bits. The learning algorithm $T$ is then used to train a model $p_{\theta_1} = T(D_1)$ on this single observation. Because the model is trained on only one datapoint, it will not be very accurate; however, it should be better than a random model that has seen no data at all. Because of the relationship between probabilistic generative models and compression described in Appendix A, we can use this model to specify the next datapoint $D_2$ in a compressed form using only $-\log_2 p_{\theta_1}(D_2)$ bits. At this point, we have encoded 2 datapoints, on which we can train a new model $p_{\theta_2} = T(D_{1:2})$. Having seen more data, this model should assign a higher likelihood to a

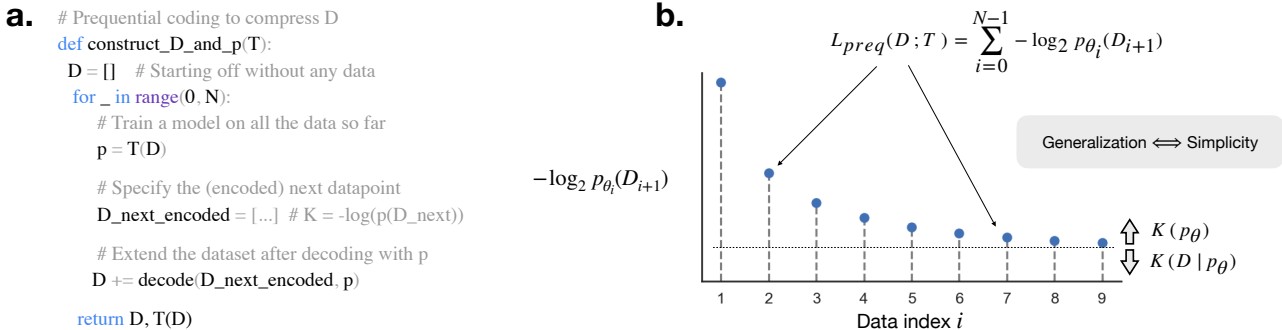

**a.**

```
# Prequential coding to compress D
def construct_D_and_p(T):
    D = []    # Starting off without any data
    for _ in range(0, N):
        # Train a model on all the data so far
        p = T(D)

        # Specify the (encoded) next datapoint
        D_next_encoded = [...]  # K = -log(p(D_next))

        # Extend the dataset after decoding with p
        D += decode(D_next_encoded, p)

    return D, T(D)
```

**b.**

$$L_{preq}(D;T) = \sum_{i=0}^{N-1} -\log_2 p_{\theta_i}(D_{i+1})$$

$-\log_2 p_{\theta_i}(D_{i+1})$

Generalization $\iff$ Simplicity

$\Uparrow K(p_\theta)$
$\Downarrow K(D \mid p_\theta)$

Data index $i$

*Figure H.1.* **Illustration of prequential coding, a method for estimating $K(D,\theta) = K(D|p_\theta) + K(p_\theta)$ using $p_\theta$'s learning algorithm $T$.**
**a.** Pseudocode of the prequential coding program that outputs both $D$ and $p_\theta$. The program jointly compresses $D$ and $p_\theta$ by incrementally training a model using $T$ on increasingly more data, each time efficiently encoding the next datapoint using the model obtained from all previous ones. The primary sources contributing to total program length come from specifying each next datapoint $D_{i+1}$ in compressed form using the current model $p_{\theta_i}$, which takes $-\log_2 p_{\theta_i}(D_{i+1})$ bits. **b.** A visual illustration of the number of bits needed to specify each next datapoint given the model that was trained on all previous ones. As the learner $T$ sees more data, it outputs models that assign a higher likelihood to new observations, and can thus better compress them. The total prequential code length $L_{preq}(D;T)$ is given by the area under the curve. The area underneath the curve's last point is equal to the number of bits needed to encode the entire dataset given the final model, $K(D|p_\theta)$. Since $L_{preq}(D;T) = K(D|p_\theta) + K(p_\theta)$, the area above the curve's last point is equal to $K(p_\theta)$. Prequential coding formalizes the intuition that simple models generalize better, thus quickly decreasing their prediction error for the next datapoint.

new datapoint $D_3$, which we can specify in compressed form using $-\log_2 p_{\theta_2}(D_3)$ bits. This process repeats until the entire dataset has been generated. At this point, the model $p_\theta$ can be obtained simply by applying the learning algorithm to the complete dataset $p_\theta = T(D)$, since we assumed by construction that this was where the model came from.

The total number of bits that it takes to jointly compress $D$ and $p_\theta$ using prequential coding is the sum of how many bits it takes to specify each next datapoint using a model that was trained on all previous ones. Visually, it is the area under the *prequential coding curve* shown in Figure H.1b. We can call the total length of this compression program the *prequential code length* $L_{preq}(D;T)$ (Blier & Ollivier, 2018):

$$L_{preq}(D;T) = \sum_{i=0}^{N-1} -\log_2 p_{\theta_i}(D_{i+1}) \tag{15}$$

$$L_{preq}(D;T) \geq K(D,p_\theta) = K(D|p_\theta) + K(p_\theta). \tag{16}$$

Strictly speaking, $L_{preq}(D;T)$ is an upper-bound on $K(D,p_\theta)$: the prequential coding algorithm is *one* way to jointly compress the data and model, but it is not necessarily the optimal way. The upper-bound is tight in practice, however, if (a) the final model $p_\theta$ does a good job of compressing the data (i.e., $K(D|p_\theta) \ll K(D)$) and (b) passing data to the learner $T$ through the prequential coding algorithm is an effective strategy for compressing the model. Regarding this second point, consider how the model is obtained through prequential coding. Data is gradually transmitted to the learner $T$, with each additional datapoint requiring fewer bits to encode. If the speed of improvement in predicting the next datapoint is fast as a function of the amount of data observed, it means that the learner is effectively able to converge to the final model using only a small amount of data that takes few bits to encode, and thus that the model has low complexity. Concretely, when prequential coding is a good algorithm for jointly compressing the data and model, then $L_{preq}(D;T) \approx K(D,p_\theta)$ and the model complexity is given by (Blier & Ollivier, 2018):

$$L_{preq}(D;T) \approx K(D|p_\theta) + K(p_\theta)$$
$$K(p_\theta) \approx L_{preq}(D;T) - K(D|p_\theta). \tag{17}$$

Assuming that the model's error decreases monotonically with the size of the training dataset, $K(D|p_\theta)$ is equal to the area under the lowest point of the prequential coding curve in Figure H.1b. The area above this point is therefore the complexity of the model $K(p_\theta)$. This relates Kolmogorov complexity to intuitions about generalization in ML: the simpler a model is, the quicker it generalizes from limited amounts of training data.

# I. Synthetic representations — experimental details

## I.1. Lookup table representations

**Generating the representations**    We generated our synthetic lookup table representations $Z$ (and their ground-truth sentences $W$) according to the program summarized in Algorithm 1. In short, the program does the following:

- **Generate a lookup table:** We begin by constructing a lookup table from words (or $n$-grams) to their embeddings. This table has dimensions $(K^q, \frac{D}{M \times q})$, where $K$ is the vocabulary size, $q$ is our disentanglement factor (i.e., the size of the $n$-grams), and $D$ is the desired dimensionality of $Z$. We use the Skellam distribution to generate lookup table entries, which is a discrete approximation of a Gaussian distribution with precision $\lambda$. This discretization is necessary because a continuous distribution would cause the correction term $K(Z|W,f)$ to be infinite.

- **Sample $W$:** We generate random integer sentences uniformly with shape $(N,L)$, where $N$ represents the number of samples and $L$ denotes the number of words per sentence. Each integer in $W$ corresponds to a word from our vocabulary of size $K$.

- **Decode $W$ to get $Z$:** For each sentence $w \in W$, we perform the following steps to obtain the corresponding representation sample $z \in Z$:
    - We divide the sentence into consecutive $L/q$ subsequences, each representing an $n$-gram (or a word if $q=1$).
    - For each subsequence, we retrieve the corresponding embedding from the lookup table.
    - We concatenate these embeddings to form the complete representation sample $z$ for the sentence.

- **Add noise:** We then add Gaussian noise (discretely approximated by a Skellam distribution with mean 0 and standard deviation $r$ for the same reason as above) to the representation. This introduces stochasticity to our representations that cannot easily be modeled with discrete parts. The final representation $Z$ has shape $(N,D)$.

**Calculating the compositionality**    To compute representational compositionality $C(Z)$ according to Definition 2, we need to calculate the following terms: $K(p_w)$, $K(W|p_w)$, $K(f)$, and $K(Z|W,f)$. We show how to do this below for a lookup table representation:

- $K(p_w)$: The language $p_w$ in this case a uniform categorical distribution over integers in range $(0,K-1)$ at each sentence position $l \in \{0..(M-1)\}$, where $K$ is the vocabulary size and $M$ is the sentence length. To specify an integer $u$, we need $\log_2 u$ bits, so we have $K(p_w) = \log_2 K + \log_2 M$. There is also a complexity term associated with describing the function for the uniform distribution itself, but we ignore this because it is a small constant.

- $K(W|p_w)$: As described in Section 2, $K(W|p_w)$ is simply equal to $-\sum_{i=1}^{N} \log_2 p_w(w_i)$. To derive $p_w(w_i)$ for each sentence $w_i \in W$, we notice that each $w_i$ is composed of $L$ words, each sample from a uniform categorical distribution over $(0, K-1)$. Thus $p_w(w_i) = \frac{1}{K^M}$ for each sentence $w_i$. In total, then, $K(W|p_w) = -\sum_{i=1}^{N} \log_2 p_w(w_i) = -\sum_{j=i}^{N} \log_2 \frac{1}{K^M} = NM \log_2 K$ bits.

- $K(f)$: In this case, the function that maps sentences to their meanings is mainly composed of the lookup table, with some additional small constant complexity to describe how to use the lookup table. To describe each number $a$ in the lookup table, we need $-\log_2 p(a)$ bits, where $p$ is the PMF of the distribution these numbers were sampled from. In our case, this distribution is the Skellam distribution with a mean of 0, a standard deviation of 1, and a precision of $\lambda$. We therefore have $K(f) = -\sum_{a \in \text{lookup table}} \log_2 p(a)$. Given that the size of the lookup table is $(K^q \times \frac{D}{M/q})$, the complexity of the semantics $K(f)$ grows linearly in $D$, polynomially in $K$, and exponentially in $q$.

- $K(Z|W,f)$: This term comes from imperfect reconstructions of $Z$. It can be thought of as the number of bits needed to correct the errors in these imperfect reconstructions. In these lookup table representations, these imperfect reconstructions come from the noise added to $Z$ when it is sampled, which cannot be recovered since the lookup table does not contain it. To describe the corrections, we therefore just need to describe this noise. Each noise sample $\epsilon$ can be described using $-\log_2 q(\epsilon)$ bits where $q$ is the PMF of the distribution the noise was sampled from. In our case this is a Skellam distribution with a mean of 0, standard deviation of $r$, and precision of $\lambda$. If we let $E$ be the matrix of all noises added form $Z$, we have that $K(Z|W,f)$ is equal to $-\sum_{\epsilon \in E} \log_2 q(\epsilon)$.

---

**Algorithm 1:** Sampling $Z$ using a lookup table program

**Input:**
> number of samples $N$
> sentence length $M$
> vocabulary size $K$
> embedding dimension $D$
> disentanglement factor $q$
> quantization precision $\lambda$
> noise ratio $r$

*// Generate lookup table:*
lookup_table $\leftarrow$ skellam_sample$(\mu=0,\sigma=1,\lambda=\lambda,\text{shape}=(K^q,\frac{D}{M/q}))$

*// Sample W:*
$W \leftarrow$ random_integer$(0,K-1,\text{shape}=(N,\mathbf{M}))$

*// Decode W to get Z:*
$Z \leftarrow []$
**for each** $w$ **in** $W$ **do**
> $z \leftarrow []$
> **for** position $=0$ **to** $(M/q)-1$ **do**
> > entry $\leftarrow (w[\text{position}\times q:\text{position}\times q+q-1])$
> > $z$.append(self.lookup_table[entry])
> **end for**
> $z \leftarrow$ concatenate$(z)$
> $Z$.append$(z)$
**end for**
$Z \leftarrow$ stack$(Z)$

*// Add noise:*
**if** $r>0$ **then**
> noise $\leftarrow$ skellam_sample$(\mu=0,\sigma=r,\lambda=\lambda,\text{shape}=Z.\text{shape})$
> $Z \leftarrow Z+$ noise
**end if**
**return** $Z$

---

Combining these complexity terms together, the final expression for $C(Z)$ following [Definition 2](#) is:

$$C(Z)=\frac{K(Z)}{K(Z|W)}=\frac{K(p_w)+K(W|p_w)+K(f)+K(Z|W,f)}{K(f)+K(Z|W,f)}$$

$$=\frac{\log_2 K+\log_2 M+NM\log_2 K-\sum_{a\in\text{lookup table}}\log_2 p(a)-\sum_{\epsilon\in E}\log_2 q(\epsilon)}{-\sum_{a\in\text{lookup table}}\log_2 p(a)-\sum_{\epsilon\in E}\log_2 q(\epsilon)}$$

**Experiment parameters**    We used the following parameter values to generate representations (except when sweeping one parameter while keeping the others constant): $N=1000$, $M=16$, $K=10$, $D=64$, $q=1$, $\lambda=0.01$, $r=0.01$. To sweep over sentence length, we varied $M$ from $(1,D)$, only keeping values where $D$ was divisible by $M$. To sweep over vocabulary size, we varied $K$ from $(2,100)$. To sweep over representation dimensionality, we varied $D$ from $(M,2M,...,10M)$. To sweep over disentanglement, we varied $q$ from $(1,M)$, only keeping values where $M$ was divisible by $q$. For each setting of experiment parameters, we generated representations across 10 different random seeds.

### I.2. Context-free grammar representations

**Generating the representations**   We generated our context-free grammar representations $Z$ (and their ground-truth sentences $W$) according to the following procedure:

- **Generate a context-free grammar:** Our context-free grammars consist of exclusively binary production rules that combine two child non-terminals into a parent non-terminal. We define a vocabulary of size $K$ and evenly assign each word to one of $T$ possible base part of speech types that serve as the first non-terminal symbols in the context-free grammar. We call these $T$ first non-terminals "terminal parts of speech". We algorithmically generate the grammar in a way that depends on two parameters: the `width` and the `depth`. The `depth` refers to the number of levels in the parse tree (above the parts of speech) that have unique non-terminal symbols which can only exist at that level. The `width` refers to the number of unique non-terminal symbols defined at each level of depth. At any given level of depth, we generate a production rule for all possible combinations of non-terminals at that level, each of which produces one of the possible non-terminals at the next level (we evenly distribute outputs across these possible non-terminals at the higher level). For arbitrarily long sentences to still have valid parses despite the finite depth of our grammar, we define additional recursive production rules that take non-terminals at the highest level of the grammar and produce one of those same non-terminals. To provide additional clarity for how we generated these grammars, we give an example below for $T = 5$, `width` $= 2$, and `depth` $= 5$ (we exclude the vocabulary for brevity). In this grammar, the terminal parts of speech are denote by the prefix "T_" and other non-terminals are denoted by the prefix "r[depth level]_".

```
s t a r t :  r2_1  |  r2_2
r0_1 :  T_1 " " T_2 | T_2 " " T_3
     |  T_3 " " T_4 | T_4 " " T_5 | T_5 " " T_1
r0_2 :  T_1 " " T_3 | T_2 " " T_4
     |  T_3 " " T_5 | T_4 " " T_1 | T_5 " " T_2
r1_1 :  r0_1 " " r0_1 | r0_2 " " r0_1
r1_2 :  r0_1 " " r0_2 | r0_2 " " r0_2
r2_1 :  r1_1 " " r1_1 | r1_2 " " r1_1
     |  r2_1 " " r2_1 | r2_2 " " r2_1
r2_2 :  r1_1 " " r1_2 | r1_2 " " r1_2
     |  r2_1 " " r2_2 | r2_2 " " r2_2
```

- **Sample $W$:** We generate random integer sentences of length $M$ based on a transmission sentence defined over terminal parts of speech. Denote a terminal part of speech by $t \in 1..T$. A sentence $w$ always randomly starts from a word that has either $t = 1$ or $t = 2$ with equal probability. Permissible transitions to the next word's terminal part of speech are $t_{i+1} \leftarrow t_i + 1$ or $t_{i+1} \leftarrow t_i + 2$, which we sample between with equal probability (we also wrap $t_{i+1}$ so that it remains in range $1..T$). Given a sampled terminal part of speech at a location in $w$, we randomly sample a word that has been assigned that terminal part of speech.

- **Semantics $f$:** The representation is assigned a dimensionality $D$. Each word in the vocabulary is given a $D$-dimensional embedding by sampling from a Skellam distribution, which is a discrete approximation of a Gaussian distribution, using $\mu = 0$, $\sigma = 1$, and quantization precision $\lambda$. For each production rule $i$ in the grammar, we define a linear mapping $A_i \in \mathbb{R}^{2D \times D}$ with values sampled from a Skellam distribution using $\mu = 0$, $\sigma = 1$, and quantization precision $\lambda$. Given a sentence $w$, the semantics function $f$ is defined by the following steps:

  - Parse $w$ using Earley parser (Earley, 1970) implemented with the `Lark` Python package.
  - Retrieve the embedding for each word in $w$.
  - Hierarchically apply the function $[x_1, x_2]A_i$ at each node in the parse tree to obtain a node embedding, where $[x_1, x_2]$ are the concatenated embeddings of the child nodes and $A_i$ is the linear transform of the production rule at the node. The embedding of the root node is taken to be $z$ for the sentence.

- **Add noise:** We then add Gaussian noise (discretely approximated by a Skellam distribution with mean 0 and standard deviation $r$) to the representation. This introduces stochasticity to our representations that cannot easily be modeled with discrete parts. The final representation $Z$ has shape $(N, D)$.

**Calculating the compositionality**  To compute representational compositionality $C(Z)$ according to Definition 2, we need to calculate the following terms: $K(p_w)$, $K(W|p_w)$, $K(f)$, and $K(Z|W,f)$. We show how to do this below for a context-free grammar representation:

- $K(p_w)$: The language $p_w$ in this case is defined by a terminal part of speech for each vocabulary item and a binary matrix of permissible transitions between terminal parts of speech. Defining the terminal part of speech for each vocabulary item takes $\log_2 T$ bits, and we have $K$ vocabulary items. The binary transition matrix is of shape $(T+1) \times T$ (where the $+1$ is for the grammar's `start` symbol), and so takes $T(T+1)$ bits to define. The total Kolmogorov complexity of the language (ignoring code of a constant complexity that doesn't scale with $K$ or $T$) is therefore $K(p_w) = K\log_2 T + T(T+1)$.

- $K(W|p_w)$: As described in Section 2, $K(W|p_w)$ is simply equal to $-\sum_{i=1}^{N}\log_2 p_w(w_i)$. Since $p_w$ is defined by a transition matrix over terminal parts of speech, and for each terminal part of speech each word having that terminal part of speech has equal probability, we have that $p_w(w_i) = \prod_{m=1}^{M}\frac{1}{|t(w_{i,m-1})|}$ where $t(\cdot)$ is the set of all permissible next words $w_{i,m}$ that the previous word $w_{i,m-1}$ can lead to based on the transition matrix between terminal parts of speech, and $w_{i,0}$ denotes the grammar's `start` symbol. We therefore have that $K(W|p_w) = -\sum_{i=1}^{N}\log_2 p_w(w_i) = -\sum_{j=i}^{N}\sum_{m=1}^{M}\log_2\frac{1}{|t(w_{i,m-1})|}$ bits.

- $K(f)$: The semantics are defined by the parser, the production rule operations (linear maps), and the word embeddings. Both the parsing algorithm and the production rule operations scale in complexity as a function of the number of production rules in the grammar, so we ignore the parsing algorithm's complexity and only consider the production rules and word embeddings as the scaling behaviour is the same. To describe each number in the word embedding table $a$, we need $-\log_2 p(a)$ bits, where $p$ is the PMF of the distribution these numbers were sampled from. In our case, this distribution is the Skellam distribution with a mean of 0, a standard deviation of 1, and a precision of $\lambda$. The complexity of the embedding table is therefore $-\sum_{a\in\text{embedding table}}\log_2 p(a)$. Given that the size of the embedding table is $(K \times D))$, the complexity of the embedding table grows linearly in both $K$ and $D$. To describe each production rule $i$, we must describe a matrix of shape $2D \times D$. Each number in this matrix takes $-\log_2 p(v)$ bits to encode, where $p$ is the PMF of the distribution these numbers were sampled from. In our case, this distribution is the Skellam distribution with a mean of 0, a standard deviation of 1, and a precision of $\lambda$. The total complexity of all production rules is therefore $-\sum_{i\in\text{num rules}}\sum_{(r,c)\in 2D\times D}\log_2 p(A_{i,(r,c)})$. We therefore have that $K(f) = -\sum_{a\in\text{embedding table}}\log_2 p(a) - \sum_{i\in\text{num rules}}\sum_{(r,c)\in 2D\times D}\log_2 p(A_{i,(r,c)})$ bits.

- $K(Z|W,f)$: This term comes from imperfect reconstructions of $Z$. It can be thought of as the number of bits needed to correct the errors in these imperfect reconstructions. In these lookup table representations, these imperfect reconstructions come from the noise added to $Z$ when it is sampled, which cannot be recovered since the lookup table does not contain it. To describe the corrections, we therefore just need to describe this noise. Each noise sample $\epsilon$ can be described using $-\log_2 q(\epsilon)$ bits where $q$ is the PMF of the distribution the noise was sampled from. In our case this is a Skellam distribution with a mean of 0, standard deviation of $r$, and precision of $\lambda$. If we let $E$ be the matrix of all noises added form $Z$, we have that $K(Z|W,f)$ is equal to $-\sum_{\epsilon\in E}\log_2 q(\epsilon)$.

Combining these complexity terms together, the final expression for $C(Z)$ following Definition 2 is:

$$
\begin{aligned}
C(Z) &= \frac{K(Z)}{K(Z|W)} = \frac{K(p_w) + K(W|p_w) + K(f) + K(Z|W,f)}{K(f) + K(Z|W,f)} \\
&= \frac{\substack{K\log_2 T + T(T+1) - \sum_{j=i}^{N}\sum_{m=1}^{M}\log_2\frac{1}{|t(w_{i,m-1})|} \\ -\sum_{a\in\text{embedding table}}\log_2 p(a) - \sum_{i\in\text{num rules}}\sum_{(r,c)\in 2D\times D}\log_2 p(A_{i,(r,c)}) - \sum_{\epsilon\in E}\log_2 q(\epsilon)}}{-\sum_{a\in\text{embedding table}}\log_2 p(a) - \sum_{i\in\text{num rules}}\sum_{(r,c)\in 2D\times D}\log_2 p(A_{i,(r,c)}) - \sum_{\epsilon\in E}\log_2 q(\epsilon)}
\end{aligned}
$$

**Experiment parameters**  We used the following parameter values to generate representations (except when sweeping one parameter while keeping the others constant): $N = 1000$, $M = 16$, $K = 100$, $D = 10$, $T = 5$, `width` $= 3$, `depth` $= 2$, $\lambda = 0.01$, $r = 0.01$. To sweep over sentence length, we varied $M$ from $(1,D)$, only keeping values where $D$ was divisible by $M$. To sweep over grammar width, we varied `width` from $(1,4)$. To sweep over grammar depth, we varied `depth` from $(1,4)$. For each setting of experiment parameters, we generated representations across 10 different random seeds.

## J. Emergent languages — experimental details

**Dataset construction** To obtain emergent languages from multi-agent reinforcement learning in a simple object reference game, both with and without iterated learning, we used the code base from Ren et al. (2020), found at `https://github.com/Joshua-Ren/Neural_Iterated_Learning`. Objects consisted of 2 attributes with 8 possible discrete values each, for a total of $8^2 = 64$ possible objects. Sentences similarly were of length 2 and had a vocabulary size of 8. We used the default values in Ren et al. (2020) for all model and training hyperparameters (refer to their associated code base for details), but reserved no held-out objects for separate validation. After training, we generated 50 sentences from the speaker agent for each unique object, giving us $W^L$ and $Z$, respectively. The resulting size of these datasets were thus $50 \times 8^2 = 3200$.

**Estimating compositionality** Estimating the compositionalities of these different emergent language systems $C^L(Z)$ requires estimates of the numerator $K(Z)$ and denominator $K(Z|W^L)$. Both with and without iterated learning, $Z$ consisted of the same enumeration over all possible discrete symbolic objects $\mathcal{O}$. Each $z \in Z$ can therefore be represented using a single integer indexing the object, where these integers range from $\{1..|\mathcal{O}|\}$ and therefore each require $\log_2(|\mathcal{O}|)$ bits to encode. Summing these bits over all objects gives a total of $K(Z) = |\mathcal{O}|\log_2(|\mathcal{O}|)$.

We estimated $K(Z|W^L)$ for each language using prequential coding (see Appendix H). The model architecture used for prequential coding was an MLP with 2 hidden layers of size 256. Each word in $W^L$ embedded into a 64-dimensional vector, and these concatenated embeddings were the input to the MLP. The MLP output logits over object values for each attribute. To estimate prequential code lengths more efficiently and avoid having to retrain the model $N$ times (where $N$ is the dataset size), we incremented the size of the dataset by chunks of size 50 at a time. We used the Adam optimizer with a learning rate of $1 \times 10^{-3}$ to train the model at each iteration of prequential coding. We reserved 400 datapoints for a separate validation set that was used for early stopping at each iteration of prequential coding.

## K. Natural languages — experimental details

**Dataset construction** We obtained English sentences from captions that were used to describe images in the Common Objects in Context (COCO) dataset (COCO, 2024), downloaded from Hugging Face. The reason for using a dataset of image captions was that we expected these captions to use common words and simple sentence structures, given their grounding in visual stimuli. For each image, the dataset contained two independent captions, and we kept only the first. This gave us a total of $414,010$ English sentences. We then translated each sentence to French, Spanish, German, and Japanese using a large open-source language model with 3.3 billion parameters (Costa-jussà et al., 2022). We visually inspected several of the French, German, and Japanese sentences (no authors spoke Spanish) to make sure the translations were reasonable, and we found them to be of high quality. These sentences constituted the $W^L$'s for our experiments. We obtained proxies for the "meanings" $Z$ of these sentences by passing them through a large (278 million parameter), pretrained, multilingual sentence embedding model that output a fixed-size vector for each sentence (Reimers & Gurevych, 2020). Both the translation model and the sentence embedding model were obtained from Hugging Face.

**Estimating compositionality** Estimating the compositionalities of these different language systems $C^L(Z)$ requires estimates of the numerator $K(Z)$ and denominator $K(Z|W^L)$. While we did not estimate $K(Z)$, we assumed that it was approximately equal among languages. This is a common assumption in linguistics, where languages appear to be equivalent in their expressive power to express ideas, refer to objects, etc. Fixing the numerator $K(Z)$ to some (unknown) constant shared among languages allowed us to assess their *relative* compositionalities by estimating only the denominator $K(Z|W^L)$. We estimated $K(Z|W^L)$ for each language using prequential coding (see Appendix H).

The model architecture used for prequential coding was the same as the one used to generate $Z$ (Reimers & Gurevych, 2020). Learning a significant number of word embeddings from only $\approx 400,000$ samples would have been difficult however. We therefore used the original model's pretrained word embeddings and only computed prequential code length by resets of the model's downstream weights, which encode the semantics of the grammar rather than the word meanings. Strictly speaking, then, we only estimated $K(Z|embeddings(W^L))$. To estimate prequential code lengths more efficiently and avoid having to retrain the model $\approx 400,000$ times, we incremented the size of the dataset in chunks. Chunk boundaries were selected on a base-10 logarithmic scale from $1,000$ to $N$ datapoints (the full size of the dataset), with 15 interval boundaries. A logarithmic scale was used because we observed that next-datapoint prediction error as a function of dataset size changed more quickly in low-data regimes and more slowly in high-data regimes. We could therefore more accurately estimate the true prequential coding curve using a logarithmic chunking scale that had higher resolution in low-data regimes. We used the Adam optimizer

with a learning rate of $1 \times 10^{-4}$ to train the model at each iteration of prequential coding. We reserved $10,000$ datapoints for a separate validation set that was used for early stopping at each iteration of prequential coding.

**Limitations**     Our approach for measuring the compositionalities of real-world language systems has several limitations that should be taken into account when judging the results. First, the translation model that we used may not have been trained on equal amounts of text from the different languages we studied, which could have lead to lower quality translations for some languages compared to others. Similarly, the multilingual sentence embedding model that we used may have not been trained on equal amounts of data from the different languages, leading to lower quality embeddings for some languages compared to others which could have impacted the quantity and accuracy of "true" sentence meaning captured in $Z$. Indeed, for these reasons we did not include the original English language sentences and embeddings in our experiments (we thought it very likely that the sentence embedding model had been trained on far more English text compared to other languages). Finally, the use of pretrained sentence embeddings as a proxy for sentence meaning $Z$ is likely flawed. The sentence embedding model that we used is trained with invariance-based self-supervised methods, and the resulting representations are unlikely to capture the full scope meaning that would be represented in human brains processing these sentences.

