# OpenReview forum: "Towards a Formal Theory of Representational Compositionality"
_ICML.cc/2025/Conference — ICML 2025 poster_

### Official Review · Reviewer_cjwu · 2025-03-08

**Overall Recommendation:** 4

**Summary:**

This paper introduces a notion of compositionality grounded in algorithmic complexity. The authors propose to treat compositionality employing Kolmogorov complexity related to representations and to a discrete language that is used to make the conversion. The contribution rests mainly in bridging how such a measure can capture compositionality, grounding many observations from intuitions in cognitive science and AI, and contrast it to topological similarity. The idea is very appealing and makes sense. The authors also present some empirical investigation of how this can be used to characterize compositionality in both synthetic and real-world representations.

**Claims And Evidence:**

The main claim is the proposal of a theory of representation compositionality that can capture intuitions from several previous works. This rests in using Kolmogorov complexity and relating representations to a sort of composition of symbols from a language. The examples and the long discussion with related literature provide evidence that Kolmogorov complexity can be useful, and overall, the idea is simple enough to adapt to different research areas. Based on the discussion and on the results, it is a particular interesting proposal.

**Essential References Not Discussed:**

N/A

**Experimental Designs Or Analyses:**

The synthetic experiments are confirming the theory. Since data are generated according to the theory, I am not particularly surprised by the representation compositionality but it is interesting to see the topological counterpart not behaving as expected. As I am not an expert in this field, I wonder if other measures of compositionality can be used in those experiments.

The algorithmic side of the paper should be more central and is now hidden in the supplementary, but it would be beneficial to know how to measure C. This can be of help with new real world experiments. Can the authors comment on the values obtained on different languages? What is a.u.? As far as I understood, C=1 reveals low compositionality and all languages reveal pretty similar representation compositionality (around 1 and 1.25)? Does this mean that the model does not attain any compositionality?

**Methods And Evaluation Criteria:**

Experiments on natural language are limited to one model and one dataset.

**Other Comments Or Suggestions:**

I suggest the authors include some examples from the supplementary material in the main text, even in a shorter version. They are quite helpful in grasping what C is measuring.

**Other Strengths And Weaknesses:**

N/A

**Questions For Authors:**

- **I have asked some questions in the previous sections.**

- **Relation to interpretability.** It would be interesting to connect this notion of compositionality and the use of language W to current focus in probing interpretability of neural/LLMs representations. Many works consider the so called "Linear Representation Hypothesis", whereby concepts (or symbols) are linearly encoded in model representations [2,3].  If these interpretable concepts are seen as elements of a language, is it possible to measure some form of representation compositionality based on that language? Do the authors have an intuition on this matter?

[2] The Linear Representation Hypothesis and the Geometry of Large Language Models, Park et al., ICML (2024) \
[3] All or None: Identifiable Linear Properties of Next-token Predictors in Language Modeling, Marconato et al., AISTATS (2025)

**Relation To Broader Scientific Literature:**

There is an interesting link to _combinatorial generalization_ [1]. In that problem, based on a limited set of observations, e.g., over variations of few object factors, the model is tested whether it generalizes to new unseen combinations of these factors. It would be interesting to see if there is a relation between this generalization and representation compositionality.

[1] The role of Disentanglement in Generalisation, Montero et al., ICLR (2021)

**Theoretical Claims:**

I found the theory pretty clear and the authors did a good work to introduce it. Some limitations are discussed by the authors, concerning the representations Z and the language W.

- **Z is continuous, W is discrete.** This is one limitation of the proposed approach. The theory of compositionality the authors propose is grounded on discrete symbols and cannot capture the complexity of real numbers. This is sensible, especially if representations capture factors of variations like the continuous values of the color of an object, its size, or whatever. It looks like this creates a complication to treat it in this framework that can only be addressed by leveraging quantization. Moreover, Kolmogorov's complexity does not easily adapt to continuous numbers.

- **The measure of compositionality requires more intuition.** I found the examples clarifying but it is not clear how values of C can be interpreted. Specifically, while C=1 is the lowest possible value, that corresponds to trivial cases, what is the meaning associated to higher values of C and is there an upper limit for C?

- **Usefulness to ood generalization and combinatorial generalization?** The idea of compositionality is especially intriguing when related to ood generalization. Is there a relation with representation compositionality? This aspect is only mentioned but it is worth expanding.

---

> ### Author Rebuttal · Authors · 2025-03-28
>
> Thank you for the constructive review.
>
> **Limitation:** $Z$ **is continuous,** $W$ **is discrete**
>
> While continuous values are problematic for Kolmogorov complexity, this may not be a significant limiting factor in practice: for instance, tokenization methods for continuous data (e.g., VQ-VAE) often exhibit surprisingly low information loss. We take the point, however, that some representation attributes (e.g., “size”) might be inherently continuous and difficult compress using a discrete $W$. We leave open the possibility that our definition can be extended to drop the requirement of a discrete $W$.
>
> **Meaning associated to higher values of** $C$
>
> It is easiest to get some intuition by fixing the denominator (which depends on $f$ and a reconstruction error) and to imagine how the numerator $K(Z)$ might be scaled. A concrete example comes from disentanglement: each word in the vocabulary has a vector embedding encoded in a lookup table of $f$, and $f$ simply concatenates or adds these vector embeddings. If we consider representations modeled by increasingly longer sentences, $f$ remains identical but $K(Z)$ keeps increasing due to increases in $-\log_2 p_w(W)$ (longer sentences have higher entropy). This is precisely what is occurring in Fig. 2b leftmost plot. The meaning of a higher $C$ is therefore that expressivity (loosely, the number of different things that can be represented) increases, but the semantics $f$ of how things are represented through parts stays the same. In theory, $C$ could be unbounded, but this requires the length of sentences to approach $\infty$, which is unlikely to be an optimal compression scheme for a representation.
>
> **Relationship to OoD compositional/combinatorial generalization?**
>
> We believe there is a relationship—we discuss this in Appendix E. Testing the empirical relationship between the two is an interesting direction for future work. We also give some hypotheses about how to specify inductive biases for compositional representation (which could improve OoD compositional generalization) in Appendix F.
>
> **Algorithmic approaches for measuring** $C(Z)$ **are in Appendix B, but should be more central**
>
> Given that we did not apply the methods in Appendix B (this is a direction left for future work, which we were transparent about in the Experiment and Conclusion sections), we thought it appropriate not to include it in the main text. The current paper focuses on validating $C(Z)$ in synthetic settings where the optimal compression of $Z$ is known from a ground-truth generative model (lines 238-246 RHS), and otherwise computes $C^L(Z)$ on real data (which is easier than $C(Z)$ because $W$ is given, lines 360-365 LHS). Our methods for estimating $C^L(Z)$ are discussed in the main text, Section 4.2.
>
> **Values for $C^L(Z)$ obtained on different natural languages**
>
> > What is a.u.?
> >
>
> Apologies, this should be in the paper: a.u. stands for arbitrary units. To compute $C^L(Z)$, we need to estimate $K(Z)$ in the numerator. For the emergent languages experiment in Section 4.2 this was simple to do (lines 352-355 RHS), but for the natural languages it is difficult. Instead, we make the (commonly-held) assumption that all languages are equally expressive in their abilities to express ideas and identify referents, which translates to equal $K(Z)$ (lines 431-435 LHS). The units are therefore “arbitrary” in Fig. 4 because we use an arbitrary constant numerator in place of $K(Z)$ that is shared among the different languages. We used the $K(Z|W,f)$ obtained from German as this constant, which also explains why German has a $C^L(Z) = 1$ in these arbitrary units: it does not mean that German obtains the trivial lowest possible compositionality, because we don’t know the true $K(Z)$ for German or the other languages. While assuming that all languages have equal $K(Z)$ simplifies analysis, it is a limitation that only allows us to compare the *relative* $C^L(Z)$ for different languages without knowing their absolute values.
>
> **Suggestion: include some examples from the supplementary material in the main text to help grasp what C is measuring**
>
> Thank you for the suggestion—we will do this.
>
> **Interpretability and linear representation hypothesis**
>
> Thank you for the great question—we have indeed thought about this! Our hypothesis is that DNNs often linearly represent these concepts because this in fact *maximizes* compositionality according to our definition: $f$ is simple because it need only store and sum word embeddings. However, we might find that far more concepts are compositionally represented and interpretable if we relax the strong assumption of linearity and allow for semantics $f$ that are more flexible, yet still simple. Interesting future work inspired by our definition could for instance train a flexible DNN as $f$ to predict LLM activations from latent concepts, and then use the same methods we applied in Sections 4.2 & 4.3 to estimate $K(f)$ and compositionality.

---

> > ### Comment · Reviewer_cjwu · 2025-04-04
> >
> > Thank you for answering my questions and for clarifying my perplexities about the measure of $C$.
> >
> >  > The units are therefore “arbitrary” in Fig. 4 because we use an arbitrary constant numerator in place of $K(Z)$ that is shared among the different languages. We used the $K(Z|W,f)$ obtained from German as this constant, which also explains why German has a $C^L(Z) = 1$ in these arbitrary units: it does not mean that German obtains the trivial lowest possible compositionality, because we don’t know the true $K(Z)$ for German or the other languages. While assuming that all languages have equal $K(Z)$ simplifies analysis, it is a limitation that only allows us to compare the relative $C^L(Z)$ for different languages without knowing their absolute values.
> >
> > This is not clear to me, and it should be mentioned in the text. Can you expand on this? Instead of arbitrary units, is it the case you are considering "relative units"? As a minor stylistic note, German should appear as the first column if it is the relative term.
> >
> > This also suggests a limitation in measuring the compositionality of languages in absolute terms. Can you further elaborate on this implication? It suggests that we do not know how models encode linguistic structure in them, in an absolute sense, but that these models just achieve higher representation compositionality of Japanese compared to German.

---

> > > ### Author Response · Authors · 2025-04-04
> > >
> > > Thank you for taking the time to read our paper and our reply in-depth, and for recognizing our contribution.
> > >
> > > > [The "arbitrary units" in Fig. 4] is [still] not clear to me, and it should be mentioned in the text. Can you expand on this?
> > >
> > > We will take care to significantly clarify this point in the text. The crux of the issue is that it is difficult to estimate $K(Z)$--which appears in the numerator of the $C^L(Z)$ term we are trying to measure--for arbitrary data such as natural language representations. This would require the application of sophisticated compression methods that attempt to tightly bound $K(Z)$. While this is by no means impossible (numerous powerful compression algorithms exist in ML, both variational and prequential in nature), we instead opted to make the simplifying assumption in this current experiment that all natural languages have the same $K(Z)$. As described previously, we used the $K(Z|W)$ obtained from German as this constant numerator in $C^L(Z)$; crucially, this numerator is "arbitrary" in the sense that we don't know what the true numerator $K(Z)$ actually is. We emphasize that if the assumption of constant numerator $K(Z)$ is correct (i.e., languages are equally expressive, which linguists believe they are), then knowing this numerator would still be essential for computing the absolute score $C^L(Z)$, but we do not need to know it to compare the *relative* $C^L(Z)$ between languages.
> > >
> > > > Instead of arbitrary units, is it the case you are considering "relative units"?
> > >
> > > Yes, precisely. In fact, we will switch the naming to "relative units" in the text and figures as this is more clear.
> > >
> > > > As a minor stylistic note, German should appear as the first column if it is the relative term.
> > >
> > > Good point, we'll make that change.
> > >
> > > > This also suggests a limitation in measuring the compositionality of languages in absolute terms. Can you further elaborate on this implication?
> > >
> > > It is indeed a limitation of our particular experiment, which did not attempt to estimate the $K(Z)$ of natural languages for the sake of simplicity. It is not, however, a fundamental limitation of the measure $C^L(Z)$, since there does exist methods for estimating $K(Z)$ (i.e., practical compression algorithms).
> > >
> > > > It suggests that we do not know how models encode linguistic structure in them, in an absolute sense, but that these models just achieve higher representation compositionality of Japanese compared to German.
> > >
> > > Precisely. We do not know the expressivity with which models encode linguistic structure $K(Z)$ in the absolute sense, but assuming that this expressivity is roughly equivalent between languages as linguists believe it is, we know that the models represent Japanese text with relatively more compositional structure than German text.
> > >
> > > We hope that this discussion has clarified our results on natural language presented in Fig. 4.

---

### Official Review · Reviewer_q4Av · 2025-03-13

**Overall Recommendation:** 3

**Summary:**

The paper builds a more rigorous version of compositional generalization, compared to the ones proposed by linguists. It claims to be the first to do so, although this may be debatable. This seems to be the first serious attempt based on kolmogorov complexity and is more agnostic of the learning model achitecture than predecessors, at the cost of being hard to compute and often even to approximately bound. However the authors cover a few examples of toy model applications, and they delineate a plan for future research in order to further "concretize" this notion.

**Claims And Evidence:**

I think that the claims are mostly correct and well defended.

I don't agree that this is exactly the first attempt at rigorously defining compositionality (this sounds as an unbearably bold statement), and the authors cite a few works that have already tried general enough (in my view) approaches. However I agree that the work is new and worth reading.

As a minor point, the authors don't discuss any alternatives to Kolmogorov Complexity as a basis for quantifying compositionality. This is a gap in the justifications which could make the notion questioned in the long term. However right now, most people agree that KC is "the" canonical measure.

I feel that the representation of previous theory of compositionality is oversimplifying and therefore several statements are misrepresenting it. The paper states that "structure" is not defined in the usual definition, then goes on to immediately contradict that by hinting at the correct statement, that actually structure is defined under assumptions on sentence parsing/type/semantics.

Furthermore, even if the abstract formulas do not have many assumptions, still e.g. the core setup summarized e.g. in Fig. B.1 involves several strong assumptions without which the actual value of C(Z) is not computable or realistic (as mentioned after Fig. B.1). But the paper's abstract claims that the new quantity is (a) quantitative and (b) conceptually simple (meaning it has no strong structural assumptions, which seems to be the main difference to previous works).

It seems that there is a strong trade-off between (a) and (b), as when we want (a) we have to give up (b) and impose structural assumptions, and when we want (b) we have to give up (a). This trade-off is fine for me, but should be highlighted for transparency. I would like to know the view of the authors on whether this trade-off is an inherent problem of KC-based metrics, or of the notion of compositionality itself: can we get rid of the trade-off?

The related claim that "sentences are simply strings" also is oversimplifying, which may lead to confusion between semiology and computer science approaches. Better say "we consider sentences as simply strings". This simplification (forgetting that sentences are uttered by intelligent agents in order to communicate) is related to the simplification that KC is considered as unquestioned as a measure of compositionality. Using KC is fine and valuable, but the fact that it is a modelling simplification should be at least hinted at, at least in the part in which the paper refers to model "brains" and human interactions at several places in the document.

**Essential References Not Discussed:**

I didn't find something concrete to point out.

**Experimental Designs Or Analyses:**

See "methods and evaluation criteria".

Another point is that in the paragraph on "vocabulary size" (line 302-304 more or less) the text says there should be a comparison between sentence length and vocabulary size. But no graph highlights that scaling/comparison. It would be good to validate the sentence by some data.

**Methods And Evaluation Criteria:**

The datasets used are a bit on the toy model side.

It would be useful to have a more thorough discussion of limitations due to the hardness of computing KC in practice.

The authors show some practical improvements like using prequential coding, VQ-VAE ideas, and others, but no experiment or comparison is present to show what can help and where. Why do you decide that "prequential coding is the answer"? I can't find where the use of this method is justified in the paper.

I think that the discussion about heuristics behing the blue curves in figure 2 is the main validation of the metric, and the rest of the experiments are not very enlightening:

- There is a graph about how compositional are different languages according to this metric. How do we know that this is realistic, is there no other benchmark or consensus about e.g. Japanese being slightly more compositional than the others? In other words I don't see how the experiment about actual languages is relevant to validating this particular metric.

- It is not clear at all that TopSim is the correct counterpart of this particular complexity measure: why is that so?
Also, what does the comparison to TopSim say, in terms of evaluation?
This differential in behavior is not discussed, so it's not clear why there is a graph between metrics.
The authors limit themselves to a series of heuristic descriptions of their own metric, and no description of "what TopSim did wrong". This being the case, why did they put the graph of TopSim metric in the paper?

**Other Comments Or Suggestions:**

line 057: compressed "more easily" or just "more"?

line 162: the \mathcal N notation was not introduced, so one has to wonder for a bit and then notice that you talk about Gaussians around that point. Maybe say it's a gaussian density before using the symbol.

line 345-348: the description of "depth" is too short, I couldn't understand it fully. maybe expand a bit

**Other Strengths And Weaknesses:**

nothing comes to mind.

**Questions For Authors:**

See the parts "claims and evidence" and "methods and evaluation criteria".

**Relation To Broader Scientific Literature:**

I think that this paper will give a good starting benchmark for the line of future research hinted at by the authors.
In particular, it invites the community to improve upon the computability/approximability of abstract KC-based metrics.

**Theoretical Claims:**

I don't think that there are theorems/proofs in the mathematical sense.
I checked the calculations and they are OK.

---

> ### Author Rebuttal · Authors · 2025-03-28
>
> Thank you for the constructive review.
>
> **Tempering claims on novelty and highlighting limitations**
>
> Reviewer JsGE made a similar comment—in retrospect, we agree. While we believe that existing definitions of compositionality suffer from pitfalls that ours addresses, it is unfair to claim that ours is the first and premature to claim that it is the “true” or most useful one. We plan to edit our paper accordingly:
>
> 1. Remove premature claims in the abstract and elsewhere that frame our definition as uniquely “correct” or “the first formal definition”.
> 2. Add a new section “Comparisons to prior work” that systematically points out advantages (e.g., generality, no assumptions on the structure of $f$) and limitations. This should help provide a neutral framing of our specific contributions.
> 3. Add a new section discussing “Limitations”, including challenges the reviewer has mentioned such as:
>     1. The difficulty of estimating KC.
>     2. The need for “strong assumptions” in practice to estimate $C(Z)$ as in Fig. B.1.
>
> **Fundamental tradeoff between (a) easily estimating complexity and (b) making strong structural assumptions?**
>
> There is indeed a fundamental tradeoff. KC requires an uncomputable search over all programs, so compression schemes impose constraints on the search space: the more constraints, the smaller the search space. We will highlight this in the paper. However, we also emphasize that the advantage of an abstract definition like ours is precisely that it allows estimators to make different tradeoffs.
>
> To some extent DNNs mitigate the tradeoff: they make few structural assumptions, but are easy to train and have inductive biases for simplicity resulting in excellent compression (Wilson 2025, Goldblum 2023). This is why we advocate for using DNNs in Appendix B.
>
> **Oversimplification of the “intuitive” definition?**
>
> > The paper states that "structure" is not defined in the usual definition, [but] structure is defined under assumptions on sentence parsing/type/semantics.
> >
>
> Our point is that the intuitive definition refers to the structure of a complex expression as if it were provided. Of course this definition is alluding to grammars, but it does not specify how such grammars are inferred for a given representation. Our definition addresses this because it precisely defines “structure” as an intrinsic property of the representation through the semantics $f$ that optimally compress it.
>
> > "sentences are simply strings" also is oversimplifying […] Better say "we consider sentences as simply strings"
> >
>
> Agreed—we will change to your wording.
>
> **Alternatives to Kolmogorov complexity**
>
> Better notions of complexity might be developed, but for the moment KC is indeed “the canonical measure”. We will better justify it, especially for unfamiliar readers. We note the presence of additional introductory material for KC in the supplement.
>
> **Why use prequential coding to measure KC?**
>
> We mention it “provides good estimates in practice”. Empirically, it provides tighter bounds on the KC of DNNs than other methods (Blier 2018), and we are using DNNs in our experiments to parameterize $f$. If other compression schemes with better bounds become available, they should be used instead. We will clarify this in the text.
>
> **Validating the natural language results?**
>
> There is no consensus on which natural languages are more/less compositional, and it is in fact a longstanding debate in linguistics. The purpose of this experiment is not to validate our compositionality metric, but rather to demonstrate it as a tool to help resolve this debate. This is explained in lines 286-296 LHS. Our results suggest that these languages are roughly equally compositional (limitations in Appendix J).
>
> **Comparisons to TopSim**
>
> We compare to TopSim because it is frequently used in the literature as a measure of compositionality (we cite several works). It is a valid competing metric because, like $C^L(Z)$, it depends on the pair $(W, Z)$. While we do not investigate “what TopSim did wrong” and why in-depth, we include it simply to show how it, as a competing metric, gives more counter-intuitive results. Whenever it deviates from the results of our definition, we flag this (e.g. that it gives nonsensical results for the natural language experiment, namely that Japanese is negatively compositional).
>
> **Other comments**
>
> > line 302-304 says there should be a comparison between sentence length and vocabulary size. It would be good to validate the sentence by some data
> >
>
> We will add a new result showing these curves for different sentence lengths.
>
> > line 057: compressed "more easily" or just "more"?
> >
>
> “more”—we will change it.
>
> > line 162: the $\mathcal{N}$ notation was not introduced […] say it's a gaussian density before
> >
>
> We will do that.
>
> > line 345-348: the description of "depth" is too short
> >
>
> It is better defined in Appendix H.2 with a concrete grammar as an example. We will define it more clearly in the main text.

---

> > ### Comment · Reviewer_q4Av · 2025-04-02
> >
> > Thank you for the response, it is along the lines that I was expecting, and confirms my initial understanding of the paper.

---

> > > ### Author Response · Authors · 2025-04-04
> > >
> > > Thank you for taking the time to read our paper and our reply in-depth, and for recognizing our contribution.

---

### Official Review · Reviewer_AgWc · 2025-03-14

**Overall Recommendation:** 4

**Summary:**

This paper argues that a quantitative measure of compositionality, beyond the traditional colloquial definition, is needed for a more precise understanding of the concept. The authors propose a measure of representational compositionality based on optimal compression using Kolmogorov complexity. Specifically, the measure is defined as the ratio of the Kolmogorov complexity of the representation to the Kolmogorov complexity of a compositional function that maps the underlying structure and parts to the representation. The authors show how the intractability of Kolmogorov complexity may be approximation in practice and demonstrate through simulation studies that meaningful estimates of the proposed measure are possible. Their numerical experiments show that the measure aligns with intuitive expectations of compositionality, capturing dependencies on sentence length, vocabulary size, representational dimensionality, and disentanglement.

**Claims And Evidence:**

The authors propose a quantitative measure of representational compositionality as the ratio of the Kolmogorov complexity of a representation to the Kolmogorov complexity of the representation given a set of sentences.

While the proposed framework is well-motivated and provides an interesting perspective on compositionality through the lens of optimal compression, there are some aspects that could benefit from further clarification and support.

A key concern is that the framework seems to address the question of whether an efficient compositional code can *generate* a representation, rather than measuring *how compositional* a given representation is itself. This distinction suggests that the measure may reflect the potential for compositional generation rather than the intrinsic compositionality of a representation. To put it as a question: if we want to study compositionality, should we aim to measure the "compositionality" of $W$, $f$, or $Z$? It seems more natural to ask if a (or what) representation $W$ can be composed into an expressive $Z$ by $f$, rather than whether $Z$ can be generated by some compositional code.

The claims that the framework addresses issues with the colloquial definition of compositionality (such as expressivity, compression, and the intrinsic nature of constituent parts) are intuitive and reasonably supported by the definition. However, claims about structure-preserving maps and modularity are weakly supported and not directly justified by the definition. Crucially, none of the five claims are validated or examined through numerical experiments.

The authors state that $p_w$, $W$, and $f$ are "not free parameters" because they are intrinsic to the representation in that they best compress $Z$. However, in practice, the measure is highly sensitive and determined by the specific choice of $f$ and $W$ (even when they are the result of a training procedure). It is unclear how the measure would generalize beyond synthetic datasets with an underlying generative model — and even in those cases, it’s plausible that some choices of $f$ and $W$ could lead to better representational compositionality scores than the actual generative model itself. Or in other words, the Kolmogorov complexity remains untraceable, even when it is decomposed into data and a model.

Lastly, the claim that the absolute value of the measure is interpretable raises some questions. The authors state that the lowest possible compositional score is 1 in the emergent language experiment, representing an arbitrary mapping from sentences to representations. However, in the natural language study, German reaches a compositionality score of ~1 — which seems counterintuitive, as natural languages are typically considered compositional. This suggests that there is a strong dependency of the calculation of $K(f)$ and $K(Z∣W,f)$ and the interpretability of absolute values of C(Z).

**Essential References Not Discussed:**

N/A

**Experimental Designs Or Analyses:**

N/A

**Methods And Evaluation Criteria:**

The authors evaluate the proposed measure across four distinct tasks: a synthetic lookup table, a context-free grammar task, an emergent language task from multi-agent training, and a natural language task using a multilingual large language model (LLM). These evaluation criteria are reasonable for demonstrating the versatility of the proposed measure across different domains. The authors show that the measure aligns with intuitive expectations of compositionality, capturing relations between compositionality and factors such as sentence length, vocabulary size, representational dimensionality, and disentanglement. Comparing the measure to topological similarity also seems reasonable, as it provides a useful benchmark for assessing structural relationships between representations.

No simulations are provided to justify expressivity, compression, the intrinsic nature of constituent, the structure-preserving maps and modularity statments.

No simulations or benchmarks are provided to show the limitations of the measure, explitly providing insights when it fails, when it fails our intuitions and when the interpretability of absolute values fails (see above).

**Other Comments Or Suggestions:**

The following sentence is difficult to follow and could benefit from a rewrite: "But where do these expressions and their constituent parts come from when considering neural representations themselves such as in the Language of Thought hypothesis, where thoughts are encoded in distributed patterns of neural activity?"

The idiom "he kicked the bucket" may be less familiar to some readers; consider using a more widely known example.

The sequence should consistently follow the same order, i.e., either K(p) + K(X|p) or K(X|p) + K(p)

Typo in section 4: "we will first illustrate [...] where where [...]
Grammar: (e.g., that it is linear, a hierarchical, etc.)

**Other Strengths And Weaknesses:**

Overall, the paper makes a strong case for the potential of the proposed measure. However, it would greatly benefit from a dedicated Limitations section that clearly and explicitly outlines the limitations of the proposed measure and highlights potential pitfalls through additional simulation studies. This would provide a more balanced perspective and help clarify the conditions under which the measure is / is not reliable and interpretable.

**Questions For Authors:**

1. In practice, do we want to measure the "compositionality" of $W$, $f$, or $Z$? It seems more natural to ask how a given representation $W$ can be composed into a meaningful $Z$ with $f$, rather than whether $Z$ can be generated by some optimal compositional code. It seems like there might be a distinction between measuring compositionality of the representation versus the process that generates it. Could you shed some light on this?
2. How, why, and when do you expect the measure to fail or produce misleading results?
3. In Section 4, what do you mean by "disentanglement," and how is it measured?
4. Could you clarify the interpretation of a compositionality score. When does it lead to an absolute value that can be interpreted and when not?
5. The claims about structure-preserving maps and modularity are not directly justified by the definition nor evaluated numerically. Do you have plans to study these aspects more explicitly?

**Relation To Broader Scientific Literature:**

Compositionality, the idea that complex expressions derive meaning from their parts and structure, has it's roots in cognitive science and linguistics. Chomsky’s theories on language productivity (Chomsky, 1956) and Fodor’s Language of Thought hypothesis (Fodor, 1975) highlight the systematic nature of thought and language, however, without providing a definition of compositionality that goes beyond intuition (Szabo, 2022). A quantitive theory of compositionality that goes beyond symbolic algorithms and provides a quantitative measure is in particular important as artificial neural networks like LLMs and brains capture and process compositional structures through abstract, non-symbolic representations. The presented paper builds on these foundations by proposing a Kolmogorov complexity-based measure of representational compositionality, linking insights from cognitive science, AI, and neuroscience.

**Theoretical Claims:**

N/A

---

> ### Author Rebuttal · Authors · 2025-03-28
>
> Thank you for the constructive review.
>
> **Q1 / Should we aim to measure the compositionality of** $W$**,** $f$**, or** $Z$**?**
>
> In our definition $C(Z)$, we are interested in the compositionality of representation $Z$ where no $W$ or $f$ are provided. We believe that our definition does in fact measure $Z$’s compositionality in terms of whether it can be expressed as a simple function of parts. We would reframe your statement as “our framework addresses the question of whether there exists an efficient code $W$ and simple model $f$ that can *generate* a given representation $Z$”, and we have argued in our paper that this is precisely how the compositionality of $Z$ should be defined.
>
> > It seems more natural to ask if a (or what) representation $W$ can be composed into an expressive $Z$ by $f$, rather than whether $Z$ can be generated by some compositional code
> >
>
> This is an interesting and related question! Whether $Z$ can be compressed through $W$ and $f$ measures compositionality, but whether there exists *other* codes $W'$ that can be composed into an expressive $Z'$ by $f$ is related to “productivity” in cognitive science (the ability to generate novel and meaningful representations using the same semantics) and compositional generalization in AI. We discuss this in Appendix E.
>
> **Q5 / Claims about structure-preserving maps and modularity are not supported**
>
> While we were not explicit about this, both were tested experimentally in Section 4.1. We will clarify in revisions.
>
> Lookup table semantics with disentanglement=1 are structure-preserving maps, and disentanglement>1 increasingly warps structure. This is why topological similarity, which only tests whether $f$ is a structure preserving map, agrees with our definition for the disentanglement results in Fig. 2b rightmost plot.
>
> The context-free grammars construct representations with modular semantics because every production rule is a separate module and each $z$ is constructed through a composition of these modules (Fig. 2c). As expected, increasing the number of modules decreases compositionality as the semantics become too complex (Fig. 2d rightmost plot), unlike highly compositiona languages in which a small number of grammar rules provide immense expressivity.
>
> > none of the five claims are validated or examined through numerical experiments
> >
>
> The other 3 claims are simply restatements of our definition:
>
> - Expressivity and compression: the numerator is expressivity, the denominator is compression with respect to parts
> - Constituent parts are intrinsic to $Z$: $W$ is defined through $Z$ in terms of optimal compression
> - Systematicity and generalization: functions with low complexity generalize better, and compositionality is maximized by low $K(f)$ in the denominator
>
> **Sensitivity to modeling choices**
>
> In our definition $f$ and $W$ are intrinsic to $Z$, but in practice modeling choices must indeed be made for these components and the sensitivity of our measure should be tested. This is a valid criticism, and in our revised paper we will have a dedicated section acknowledging limitations to be addressed in future work (following your suggestion to include such a section).
>
> We note, however, that the abstract definition can be useful in and of itself. Kolmogorov complexity for instance is uncomputable but still conceptually useful.
>
> **Q4 / German reaches a minimal compositionality score of ~1, which seems to invalidate claims about interpretable absolute values**
>
> This is a lack of clarity on our part—German does not have a compositionality of 1. In Fig. 4, a.u. stands for arbitrary units. To compute $C^L(Z)$, we need to estimate $K(Z)$ in the numerator. For the emergent languages this is simple to do (lines 352-355 RHS), but for natural languages it is not. Instead, we make the commonly-held assumption that all languages are equally expressive, which translates to equal $K(Z)$ (lines 431-435 LHS). The units are therefore “arbitrary” because we use an arbitrary constant numerator in place of $K(Z)$ that is shared for all languages. We use the $K(Z|W,f)$ obtained from German as this constant, which also explains why German has a $C^L(Z) = 1$ in these arbitrary units. While assuming equal $K(Z)$ simplifies analysis, it is a limitation that only allows us to compare the *relative* $C^L(Z)$ for different languages without knowing their absolute values—we will clarify this in the text.
>
> **Q2**
>
> We expect the definition to fail when implemented with improper modeling assumptions (e.g., wrong DNN architecture for $f$), poor training hyperparameters, or insufficient data for training DNNs. We will expand on this in a new Limitations section.
>
> **Q3**
>
> Disentanglement (defined in lines 248-258 RHS) refers to the size of the n-grams used to generate our synthetic lookup table representations. For instance, if disentanglement=2, the lookup table has an entry for all possible *pairs* of words and $z$ is generated by concatenating these pair embeddings.

---

> > ### Comment · Reviewer_AgWc · 2025-04-02
> >
> > I appreciate the authors’ thoughtful and detailed response in addressing the concerns I raised. Their clarifications have been valuable in better understanding the scope and limitations of the proposed approach.
> >
> > I also appreciate that the authors recognize the need to temper some of the claims in general (see also their responses to reviewers JsGE and q4Av) and to more precisely reflect the conditions under which they hold. In particular, the definition heavily dependents on modeling choices. Additionally, claims regarding structure-preserving maps and modularity seem to be supported primarily in highly controlled toy setups, rather than in more complex or real-world scenarios. Similarly, while the interpretability of the compositionality score holds in controlled settings, it does not generalize well, as evidenced by its breakdown in the natural language experiments ...
> >
> > That said, I fully agree that "the abstract definition can be useful in and of itself," and in light of these clarifications, I have decided to revise my overall recommendation.

---

> > > ### Author Response · Authors · 2025-04-04
> > >
> > > Thank you for taking the time to read our paper and our reply in-depth, and for recognizing our contribution.
> > >
> > > One minor clarification we would like to make: we don't believe that the interpretability of the compositionality score breaks down in our natural language experiments. The absolute value of the score no longer has meaning, but it is still valuable for assessing the *relative* compositionalities of natural languages and gives interpretable results (unlike an alternative metric like topological similarity, which is also a relative score but gives counter-intuitive results in our natural language experiments).

---

### Official Review · Reviewer_JsGE · 2025-03-22

**Overall Recommendation:** 2

**Summary:**

This submission frames compositionality as a quantitative measure of how compressible a representation is into a specific family of probabilistic models. This quantitative measure of compositionality is tested in three settings: One in which the generative model of the data is known and specific parameters can be controlled, and two in which it is not known.

**Claims And Evidence:**

The abstract boldly claims that: "Our definition has the potential to inspire the design of novel, theoretically-driven models that better capture the mechanisms of compositional thought." This is quite a bold claim for a paper that has three small experiments, one on procedurally generated data. I believe a better description of this paper is: a model-based quantitative measure of compositionality.

**Essential References Not Discussed:**

Several important works in cognitive science:

1. Cognition as compression in a probabilistic framework, efficient coding:

    Chater N, Vitányi P. Simplicity: a unifying principle in cognitive science? Trends Cogn Sci. 2003 Jan;7(1):19-22. doi: 10.1016/s1364-6613(02)00005-0. PMID: 12517354.

    Feldman J. The simplicity principle in perception and cognition. Wiley Interdiscip Rev Cogn Sci. 2016 Sep;7(5):330-40. doi: 10.1002/wcs.1406. Epub 2016 Jul 29. PMID: 27470193; PMCID: PMC5125387.

And a methodological connection in computer science:

1. Connections between compression and induction of simple grammars:

    Adriaans, P., Jacobs, C. (2006). Using MDL for Grammar Induction. In: Sakakibara, Y., Kobayashi, S., Sato, K., Nishino, T., Tomita, E. (eds) Grammatical Inference: Algorithms and Applications. ICGI 2006. Lecture Notes in Computer Science(), vol 4201. Springer, Berlin, Heidelberg. https://doi.org/10.1007/11872436_24

**Experimental Designs Or Analyses:**

> Figure 4. Compositionality of natural language systems.

This is not natural language data but instead elicitations from an LLM. That should be made clear.

**Methods And Evaluation Criteria:**

adequate

**Other Comments Or Suggestions:**

> Thus, in practice, Z must be discretized to some finite precision and a discrete approximation of the Normal distribution must be used (e.g., the Skellam distribution).

I think you mean "can be used" instead of "must be used". Specific approximations, including this one, are not justified a priori.

> we first collected a dataset of English sentences describing natural images (COCO, 2024)

Did you collect the dataset? Or did you make use of it?

> We introduced a novel definition of compositionality, representational compositionality, that is grounded in algorithmic information theory.

Isn't compositionality always about representations?

**Other Strengths And Weaknesses:**

#### Stengths

Very clear.

#### Weaknesses

Claims are too advanced for what is shown relative to prior work.

**Questions For Authors:**

1. Given that connections between compression and compositionality have been made by Kirby, including in emergent communication, what is the advancement that your work brings? If it is a specific methodology for approximating compositionality, and/or a drop-in replacement for  topological similarity, then the paper should be reframed to make that claim central, rather than overclaiming about a new approach to explaining compositionality in language and thought.

1. There is almost no interpretation of the "natural" language results in Figure 4 other than no significant differences per the measure of representational compositionality. What are we supposed to conclude from this experiment?

**Relation To Broader Scientific Literature:**

Realizes compositionality as compression, following closest to the work of Kirby that shows that compositional language structures emerge via cultural transmission as a balance between expressive communication and efficient (compressed) encoding (Kirby, 2015; 2019).

**Theoretical Claims:**

n/a

---

> ### Author Rebuttal · Authors · 2025-03-28
>
> Thank you for the constructive review.
>
> **Tempering our claims**
>
> Reviewer q4Av made a similar comment—in retrospect, we agree. While we believe that existing definitions of compositionality suffer from significant pitfalls which our definition addresses, it is unfair to claim that ours is the first to rigorously define compositionality and premature to claim that it is the “true” or most useful definition in all circumstances. It is also premature to claim that it has the potential to dramatically improve AI models, and such claims are best left for future work that actually attempts this. We plan to edit our paper accordingly in the following ways:
>
> 1. Remove premature claims in the abstract and elsewhere that frame the definition as uniquely “correct”, being “the first formal definition”, or having the potential to dramatically improve AI.
> 2. Add a new section “Comparisons to prior work” that systematically compares our work to other definitions of compositionality that have been proposed, pointing out both advantages (e.g., generality, no assumptions on the structure of $f$) and limitations (e.g., computability, sensitivity to modeling choices in practice). This should help provide a neutral framing of our specific contributions.
> 3. Add a new section discussing additional “Limitations” of our definition.
>
> **Discussing additional references**
>
> The Chater and Feldman references are only tangentially related, as we don’t define compositional representations as simple (only that they can be compressed as a simple function of parts)—i.e., $C(Z) \neq K(Z)$. However, we will add these references when discussing compression more broadly (paragraph on “expressivity and compression”).
>
> The Adriaans & Jacobs reference is highly relevant as it pertains to finding simple grammars that explain data—the function $f$ in our framework—which is required to compute $C(Z)$. Thank you! We will include this.
>
> **Advances upon Kirby et al.’s work on compositionality**
>
> 1. Our definition provides a formal and quantitative framing of ideas in Kirby’s work, clearly defining what is meant by a language (symbols $W$, meanings $Z$, and their mapping $f$), effective communication (high $K(Z)$ that can represent many things, low information loss during communication $K(Z|W, f)$), and language simplicity (low $K(f)$). Lines 196-202 RHS made this contribution explicit, but we will make this more central throughout the paper (especially in the abstract, introduction, and discussion) so that readers understand how we are building on existing literature.
> 2. Kirby’s work presents ideas around the compositionality of *language systems, or mappings from a given* $W \rightarrow Z$, which pertains to our $C^L(Z)$ definition. In this sense, $C^L(Z)$ is a drop-in replacement for topological similarity*.* While this is relevant to natural language, it is not directly applicable to questions about the compositionality of *representations, where only* $Z$ *is given*, such as in testing the Language of Thought hypothesis. Our definition $C(Z)$ (but not Kirby’s theory or topological similarity) remains applicable in this case.
>
> In our revisions, we will summarize these points.
>
> **Interpretation of natural language results**
>
> The motivation for the natural language experiments is stated on lines 386-392 LHS: it is unknown whether different languages are equally compositional, partly because we lack principled definitions of compositionality. We therefore took this as an opportunity to apply to our definition to a real problem in linguistics. Our results in Fig. 4 suggest that these natural languages are roughly equally compositional. We will edit our paper to relate these results back to the original motivation for the experiment. We also wanted to show how another measure often used as a proxy for compositionality of language systems, topological similarity, gives different and counter-intuitive results.
>
> **Other comments**
>
> > I think you mean [a discrete approximation of the Normal distribution] "can be used" instead of "must be used"
> >
>
> Yes, thank you; we will correct the text.
>
> > Did you collect the [English sentences] dataset? Or did you make use of it?
> >
>
> We made use of the existing dataset; we will correct the text.
>
> > Isn't compositionality always about representations?
> >
>
> It is discussed more broadly in the literature, or at least would require a much broader notion of “representation” than the one used in our paper. It is sometimes about data [e.g., 1], functions [e.g., 2],  or mappings from externally-defined latents to a representation as in $C^L(Z)$ [e.g., 3]. We can clarify this in the text to better situate our definition in the broader AI and cognitive science communities.
>
> [1] Aitchison 1982. The Statistical Analysis of Compositional Data
>
> [2] Lepori 2023. Break It Down: Evidence for Structural Compositionality in Neural Networks
>
> [3] Ren 2023. Improving Compositional Generalization Using Iterated Learning and Simplicial Embeddings

---

### Decision · Program_Chairs · 2025-05-01

**Decision:**

Accept (poster)

**Comment:**

This paper introduces a formal, quantitative definition of representational compositionality based on Kolmogorov complexity. The main contributions include
- It reframes compositionality as the compressibility of representations into structured functions of their parts. This definition offers a compelling alternative to heuristic or symbolic notions and aligns with intuitive expectations across multiple domains.
- The paper evaluates the proposed measure in synthetic setups, emergent language scenarios, and multilingual LLM embeddings, and contrasts it with topological similarity.
- It serves as a bridge between cognitive and computational perspectives.

During the response period, the authors addressed concerns about overclaims, interpretability, and modeling sensitivity. They committed to tempering claims, adding a limitations section, clarifying assumptions about absolute vs. relative compositionality scores, and situating their work more precisely with respect to prior definitions (e.g., Kirby mentioned by Reviewer JsGE). While Reviewer JsGE maintained a weak reject, citing limited empirical depth, other reviewers increased their scores post-rebuttal. Given the paper’s conceptual clarity, rigor, and potential for impact, I recommend acceptance.

For the final version, the authors are suggested to (1) temper novelty claims, and (2) improve clarity on practical estimation of Kolmogorov complexity.